



Atmospheric
Chemistry
and Physics

# How well do the CMIP6 models simulate dust aerosols?

**Alcide Zhao[1,2], Claire L. Ryder[1], and Laura J. Wilcox[1,2]**

[1]Department of Meteorology, University of Reading, Reading, UK
[2]National Centre for Atmospheric Science, Reading, UK

**Correspondence:** Alcide Zhao (alcide.zhao@reading.ac.uk)

**Abstract.** Mineral dust impacts key processes in the Earth system, including the radiation budget, clouds, and nutrient cycles. We evaluate dust aerosols in 16 models participating in the sixth phase of the Coupled Model Intercomparison Project (CMIP6) against multiple reanalyses and observations. We note that both the reanalyses and observations used here have their limitations and particularly that dust emission and deposition in reanalyses are poorly constrained. Most models, and particularly the multi-model ensemble mean (MEM), capture the spatial patterns and seasonal cycles of global dust processes well. However, large uncertainties and inter-model diversity are found. For example, global dust emissions, primarily driven by model-simulated surface winds, vary by a factor of 5 across models, while the MEM estimate is double the amount in reanalyses. The ranges of CMIP6 model-simulated global dust emission, deposition, burden, and optical depth (DOD) are larger than previous generations of models. Models present considerable disagreement in dust seasonal cycles over North China and North America. Here, DOD values are overestimated by most CMIP6 models, with the MEM estimate 1.2–1.7 times larger compared to satellite and reanalysis datasets. Such overestimates can reach up to a factor of 5 in individual models. Models also fail to reproduce some key features of the regional dust distribution, such as dust accumulation along the southern edge of the Himalayas. Overall, there are still large uncertainties in CMIP6 models' simulated dust processes, which feature inconsistent biases throughout the dust life cycle between models, particularly in the relationship connecting dust mass to DOD. Our results imply that modelled dust processes are becoming more uncertain as models become more sophisticated. More detailed output and dust size-resolved variables in particular, relating to the dust cycle in future intercomparison projects, are needed to enable better constraints of global dust cycles and enable the potential identification of observationally constrained links between dust cycles and optical properties.

## 1   Introduction

Mineral dust, a key component of the Earth system, has important impacts on the global climate and environment through a number of pathways (Mahowald et al., 2010; Gassó et al., 2010; Knippertz and Stuut, 2014; Shao et al., 2011; Mahowald et al., 2014; Kok et al., 2018; Jin et al., 2021). For example, links have been found between dust emissions and Atlantic hurricanes, Amazon forest fertilisation (Yu et al., 2015), and the African and Indian monsoons (N'Datchoh et al., 2018; Skonieczny et al., 2019; Maharana et al., 2019; Jin et al., 2021). There remain considerable gaps in our understanding of dust throughout its life cycle (i.e. emission,

transport, and deposition) due partly to challenges in dust observations (Richter and Gill, 2018), hindering complete understanding and modelling of the complex roles of dust aerosols in the Earth system.

Dust aerosols have been included in global climate and Earth system models since the late 1980s (Shao et al., 2011). These models, with increasingly finer resolutions and more sophisticated model physics and parameterisations, demonstrate certain capabilities in simulating mesoscale to global-scale dust events and processes. However, large uncertainties exist in dust simulations stemming from many sources (Evan et al., 2014; Wu et al., 2018; M. Wu et al., 2020; Adebiyi and Kok, 2020) – for example, incomplete understanding

and representations of the driving mechanisms of dust emission, transport, and deposition; dust particle size and shape; and model structural differences. It also remains a challenge for climate models to accurately simulate the meteorological processes that play critical roles in dust processes. Yet, these uncertainties tend to amplify as models become more complex (Kok et al., 2017; Ryder et al., 2019; Adebiyi and Kok, 2020; Di Biagio et al., 2020; Kramer et al., 2020; Li et al., 2021; Huang et al., 2021).

Uncertainties in the simulation of dust have important implications for interpreting the model-simulated global radiation budgets and many processes that are influenced by dust (e.g. clouds). Also, in the context of global efforts to mitigate anthropogenic aerosol and precursor emissions, natural aerosols like dust will potentially form a relatively greater and yet uncertain contribution to global aerosol concentrations in shaping future climate variability. Therefore, it is crucial to understand the performance of dust simulations in the latest-generation climate models.

The first multi-model and multi-parameter evaluation of dust simulations was carried out within 15 models participating in phase I of the Aerosol Comparisons between Observations and Models (AeroCom I) (Huneeus et al., 2011). These models were able to reproduce vertically integrated parameters such as dust aerosol optical depth (DOD) within a factor of 2 and the dust deposition and surface concentration within a factor of 10. Kim et al. (2014, 2019) evaluated AeroCom phase II model-simulated DOD over North Africa and East Asia against multiple observational datasets and found these models significantly underestimated dust transport to adjacent oceans. The latest AeroCom phase III models are reported to have better-resolved dust particle size distributions compared to those in phase I and II (Gliß et al., 2021). However, dust particles are still too fine compared to the Aerosol Robotic Network (AERONET) retrievals (Holben et al., 1998). Also, large diversities were found across different models in the simulations of dust emission, burden, and lifetime. This leads to diversities in dust spatial distributions and transport to the oceans. These have implications for interpreting the diversities in model-simulated aerosol optical properties and aerosol–radiation–cloud interactions.

Several studies have examined the performance of the CMIP5 models in dust simulations at both regional and global scales. For example, Evan et al. (2014) found that the African dust emissions and burdens were systematically underestimated in 23 CMIP5 models, while their year-to-year changes were poorly constrained compared to observations. Similarly, it was shown that CMIP5 models significantly underestimated dust transport to the Indian subcontinent because of biases in the model-simulated 850 hPa winds (Sanap et al., 2014). Wu et al. (2019) found large discrepancies between observed and CMIP5 models' simulated decadal variabilities of dust emissions over East Asia and questioned the implications for long-term variations in dust-related processes. Pu and Ginoux (2018) compared seven

CMIP5 models' simulated DOD to the Moderate Resolution Imaging Spectroradiometer (MODIS) Deep Blue aerosol product. They found that the multi-model mean was better than most individual models in capturing the climatology and seasonal cycles of DOD over most dust source regions but that it still underestimated the mean value and the amplitude of the seasonal cycle. This is consistent with the representation of wind/precipitation processes in the models (i.e. the multi-model mean outperforms individual models) (Sperber et al., 2013). Almost all the seven models failed to capture the DOD interannual variations. Dust cycles in the CMIP5 models were further evaluated by [TS1] C. Wu et al. (2020) against the MERRA2 aerosol reanalysis and station observations. They found that CMIP5 models, compared to the AeroCom II models, featured amplified model diversities and attributed this to increases in model complexities such as the coupling between dust emissions and dynamic vegetation. In short, although CMIP5 models were able to simulate some aspects of dust distribution and seasonal cycles well, their ability to represent certain features was still limited, and inter-model variability was too large to provide useful constraints on dust interactions with the climate system.

The CMIP6 models (Eyring et al., 2016) represent significant advances compared to the CMIP5 models in many ways – for example, the inclusion of additional Earth system components and processes, such as dynamic vegetation, in a greater proportion of models. For dust aerosols, given the large uncertainties in previous generations of climate models discussed above, it is important to evaluate the performance of the CMIP6 models – in particular, how well do these models simulate dust processes compared to each other and compared to observations and previous generations of models. Such understanding would serve as a benchmark for the dust-modelling community to interpret a variety of processes related to dust in climate models, while also helping climate model centres to develop their models into the next phase, and help target future observations directed towards constraining model processes.

Here we provide the first comprehensive intercomparison and evaluation of the CMIP6 models in dust simulations at the global scale while focusing on a few key dust source regions. We examine 16 CMIP6 models that performed the Atmospheric Model Intercomparison Project (AMIP) experiment to limit the influence of internal variability on inter-model diversity. We compare model-simulated dust emission, deposition, burden, lifetime, and DOD to multiple reanalyses and observational datasets. We also examine the driving processes of dust emissions using a regression technique. This paper is organised as follows. Section 2 briefly introduces the 16 models and simulations we examine in this work, as well as the reanalysis and observational datasets, and statistical analyses. Results are presented in Sect. 3, followed by a summary of key findings and discussions in Sect. 4.

                                      https://doi.org/10.5194/acp-22-1-2022

## 2 Data and methods

### 2.1 CMIP6 AMIP models and simulations

We examine dust in 16 climate and Earth system models (hereafter ESMs; Table 1) participating in the CMIP6 AMIP (Eyring et al., 2016). These models were selected based on solely the criterion that at least the monthly mean DOD field was available at the time of writing. AMIP is one of the four CMIP6 baseline Diagnostic, Evaluation and Characterization of Klima (DECK) experiments. In AMIP, sea surface temperature and sea ice are prescribed from observations, so that the atmospheric and land components within each model can be evaluated under the constraint of observed ocean conditions. Subcomponent models in each ESM, as well as external forcings such as greenhouse gas concentrations and land use, are identical to those in the CMIP6 historical simulations. All models analysed here cover at least the period 1979–2014. We focus on the present-day (2005–2014) period for model evaluation, guided by the availability of observational (i.e. satellite and ground observations) and reanalysis datasets (see Sect. 2.2–2.3). In Sect. 3.2, we also use the 1985–2014 data to ensure the robustness of the regression analysis for determining dust emission drivers.

Unlike previous-generation CMIP models, dust emissions in almost all the 16 CMIP6 models (except INM-CM4-8) are calculated online and resolved into different size bins (see Table 1). However, the dust particle size range represented differs significantly between models, with the use of bin-based and model schemes, as well as maximum diameter (bin-based) ranging from 0.01 up to 63 μm in diameter. Depending on the model, dust emissions are calculated based on factors including surface winds, land surface properties, and vegetation. Dust particles interact with clouds by serving as cloud condensation nuclei in most of the 16 models; however, only two models (MRI-ESM2-0 and NorESM2-LM) have realised dust particles as ice nuclei.

For models that have more than one ensemble member, we average these members to produce a model ensemble mean unless otherwise stated. The model ensemble mean is used to represent each individual model and is interpolated to the UKESM1-0-LL model grid ($1.25° \times 1.875°$) when calculating the multi-model ensemble mean (MEM). The UKESM1-0-LL model grid was chosen as it is the intermediate horizontal resolution between the highest and the lowest ones. We also calculate the 10th–90th percentiles of the multi-model spreads when producing zonal and meridional mean profiles, as well as the seasonal cycles of regional mean dust emissions and DOD. The climatological means were calculated as averages of the 2005–2014 annual means from each model ensemble mean.

### 2.2 Satellite and ground observations

Satellite observations are one of the most reliable tools for constraining and evaluating ESMs at the global scale (Flato et al., 2013). Here we use satellite AOD and DOD retrievals at 550 nm to evaluate the performance of the CMIP6 AMIP models over the period 2005–2014.

There are currently several satellite DOD products developed using the MODIS and/or the Advanced Very High Resolution Radiometer (AVHRR) observations (Ginoux et al., 2012; Pu and Ginoux, 2018; Voss and Evan, 2020), and each has its own limitations and advantages. Here we use the ModIs Dust AeroSol (MIDAS) dataset (Gkikas et al., 2021) that provides global-scale land and ocean daily DOD with fine spatial resolution ($0.1° \times 0.1°$) for the period 2003–2016. MIDAS was calculated using quality-filtered MODIS-Aqua AOD retrievals along with DOD-to-AOD ratios provided by the Modern-Era Retrospective analysis for Research and Applications version 2 reanalysis (MERRA2). This means that the MIDAS DOD estimates are also model-dependent and uncertain. The MIDAS dataset was validated against the AERONET and the LIdar climatology of vertical Aerosol Structure for space-based lidar simulation (LIVAS) DOD products, and it was demonstrated to be suitable for DOD climatology study and model evaluation (Gkikas et al., 2021).

We also use the 12-satellite merged AOD product developed by Sogacheva et al. (2020) at the Finnish Meteorological Institute (FMI AOD hereafter). FMI AOD provides monthly data for the period 1995–2017 at a $1° \times 1°$ horizontal resolution. It has better spatial and temporal coverage than any individual satellite AOD products, while the quality of the merged product is at least as good as that of individual products.

To evaluate models' simulated dust deposition fluxes, we used the ground deposition flux of dust with a geometric diameter $\leq 10$ μm ($PM_{10}$) in around 110 stations (Fig. 1) compiled by Albani et al. (2014). Note that due to the availability of surface dust concentration from CMIP6 models, we were not able to evaluate dust concentrations against ground observations.

### 2.3 CAMS and MERRA2 reanalyses

The Copernicus Atmosphere Monitoring Service (CAMS; Inness et al., 2019) reanalysis represents the latest global reanalysis dataset of the atmospheric composition produced by the European Centre for Medium-Range Weather Forecasts (ECMWF). It assimilates satellite retrievals of many atmospheric constituents including CO, $NO_2$, $O_3$, and AOD from MODIS Terra and Aqua and AATSR (Advanced Along-Track Scanning Radiometer) Envisat, using the ECMWF's Integrated Forecasting System. The CAMS reanalysis is available from 2003 onward at a horizontal resolution of $\sim 80$ km. Dust emission is calculated based on Ginoux et

**Table 1.** Summary of models and simulations used in this study.

| No. | Models | Ensemble members | Resolution (lat × long × level) | Dust size diameter boundaries (μm) | Dust emission scheme | Model references |
|---|---|---|---|---|---|---|
| a | CESM2-CAM | 10 | 0.9° × 1.25° × 32L | Three modes: Aitken: 0.01–0.1; accumulation: 0.1–1.0; coarse: 1.0–10.0 | Zender et al. (2003) | Liu et al. (2016), Danabasoglu et al. (2020) |
| b | CESM2-CAM-FV2 | 3 | 1.9° × 2.5° × 32L | | | |
| c | CESM2-WACCM | 3 | 0.9° × 1.25° × 70L | | | |
| d | CESM2-WACCM-FV2 | 3 | 1.9° × 2.5° × 70L | | | |
| e | CNRM-ESM2-1 | 1 | 1.4° × 1.41° × 90L | 3 bins: 0.01–1.0, 1.0–2.5, 2.5–20 | Marticorena and Bergametti (1995), Kok (2011) | Séférian et al. (2019) |
| f | CanESM5 | 5 | 2.8° × 2.8° × 49L | Bulk concentrations | Marticorena and Bergametti (1995) | Swart et al. (2019) |
| g | GISS-E2-1-G | 5 | 2° × 2.5° × 40L | 6 bins: < 1, 1–2, 2–4, 4–8, | Cakmur et al. (2006), | Bauer et al. (2020) |
| h | GISS-E2-2-G | 5 | 2° × 2.5° × 102L | 8–16, 16–32 | Miller et al. (2006) | Rind et al. (2020) |
| i | HadGEM3-GC31-LL | 1 | 1.25° × 1.875° × 85L | 6 bins: 0.064–0.2, 0.2–0.63, 0.63–2.0, 2.0–6.32, 6.32–20, 20–63 | Marticorena and Bergametti (1995), Woodward (2001) | Williams et al. (2018) |
| j | INM-CM4-8 | 1 | 1.5° × 2° × 21L | Bulk concentration | Prescribed | Volodin et al. (2018) |
| k | INM-CM5-0 | 1 | 1.5° × 2° × 73L | Bulk concentration | Volodin and Kostrykin (2016) | Volodin and Gritsun (2018) |
| l | IPSL-CM6A-LR | 4 | 1.26° × 2.5° × 79L | 1 lognormal mode with mass median diameter (geometric standard deviation): 2.5 (2) | Di Biagio et al. (2020) | Boucher et al. (2020) |
| m | MIROC-ES2L | 2 | 4.5° × 2.8° × 40L | 10 bins: 0.1–0.16, 0.16–0.25, 0.25–0.40, 0.40–0.63, 0.63–1.00, 1.00–1.58, 1.58, 2.51, 2.51–3.98, 3.98–6.31, 6.31–10 | Takemura et al. (2000) | Hajima et al. (2020) |
| n | MRI-ESM2-0 | 1 | 1.125° × 1.125° × 80L | 6 bins: 0.2–2, 2–4, 4–6, 6–8, 8–10, 10–12, 12–20 | Tanaka and Chiba (2005) | Yukimoto et al. (2019) |
| o | NorESM2-LM | 1 | 1.875° × 2.5° × 32L | Same as CESM2 | Zender et al. (2003) | Seland et al. (2020) |
| p | UKESM1-0-LL | 1 | 1.25° × 1.875° × 85L | 6 bins: 0.064–0.2, 0.2–0.63, 0.63–2.0, 2.0–6.32, 6.32–20, 20–63 | Marticorena and Bergametti (1995), Woodward (2001) | Senior et al. (2020) |
| | MERRA2 | | 0.5° × 0.625° × 72L | 5 bins: 0.2–2, 2–3.6, 3.6–6, 6–12, 12–20 | Ginoux et al. (2001) | Gelaro [TS2] at al. (2017) |
| | CAMS | | 0.75° × 0.75° × 60L | 3 bins: 0.06–1.1, 1.1–1.8, 1.8–40 | Ginoux et al. (2001) | Inness et al. (2019) |

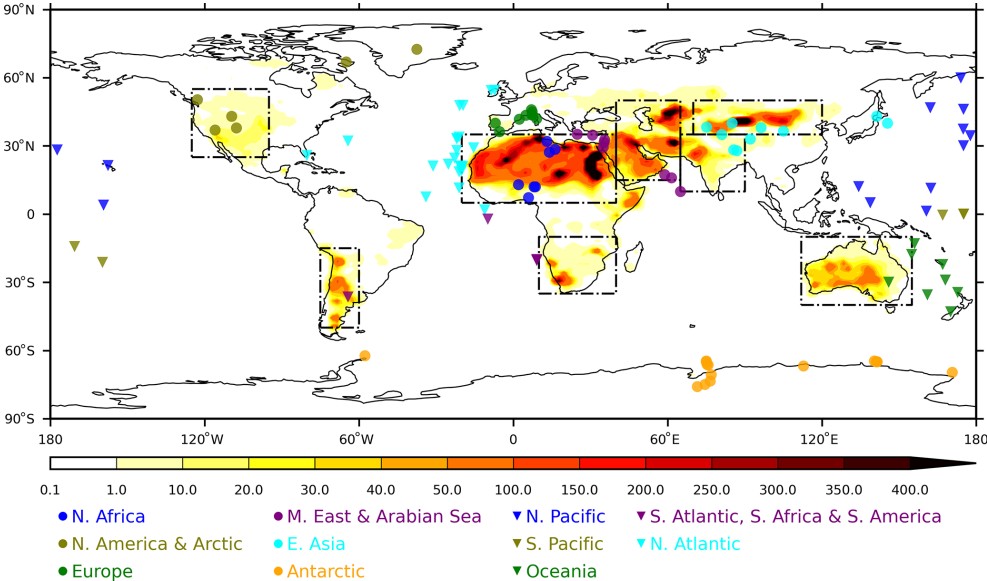

**Figure 1.** The CMIP6 AMIP MEM-simulated 2005–2014 annual mean dust emission ($\mathrm{g\,m^{-2}\,yr^{-1}}$) climatology overlaid by boxes used to define major dust emission source regions. The coloured symbols denote groupings of observations by different regions following Kok et al. (2021).

al. (2001) and is resolved in three size bins with diameter bounds at 0.06,1.1, 1.8, and 40 μm, respectively (Table 1). Monthly mean AOD and DOD are available, while dust cycle fluxes including dust emission and deposition are provided at 3 h intervals which were processed into monthly means. We also use the 2005–2014 monthly mean bare soil fraction and leaf area index from CAMS, as well as soil moisture, surface winds, and precipitation from ECMWF Reanalysis version 5 (ERA5), to investigate the drivers of dust emissions (see Sect. 2.4).

In addition to CAMS, we also use the MERRA2 reanalysis which was produced using the Goddard Earth Observing System (GEOS-5; Molod et al., 2015) with a 3D variational data assimilation system (3D-Var) that assimilates a wide range of observational datasets (Gelaro et al., 2017). For AOD, MERRA2 assimilates data from AVHRR, MODIS, the Multi-angle Imaging SpectroRadiometer (MISR), and AERONET. Dust emission is simulated based on Ginoux et al. (2001) and is resolved in five size bins with diameter bounds at 0.2, 2.0, 3.6, 6.0, 12.0, and 20.0 μm respectively (Table 1). In addition to its AOD and DOD products, MERRA2 also provides dust emission and deposition fluxes at 0.1° horizontal resolution from 1980 onward, making it a valuable tool for evaluating dust processes in climate models. We use the MERRA2 monthly mean AOD and DOD, as well as dust cycle fluxes (emission, dry deposition and wet deposition, and burden) over the period 2005–2014. Note that we were not able to investigate the drivers of dust emissions using MERRA2 because bare soil fraction data are not provided.

It is important to note that only AOD from observations is assimilated in CAMS and MERRA2. The DOD and dust mass loading are then adjusted based on the contribution of DOD to AOD, which will vary in space and time. Therefore, the accurate representation of DOD and dust mass loading in the reanalyses relies on the simulation of correct proportions of dust relative to other aerosol species. While this aerosol speciation may be well represented in locations or time periods dominated by dust (e.g. over the remote Sahara), it is likely to be less well represented in regions where different aerosol species coexist (e.g. over northern India, with mixed dust, smoke, and anthropogenic aerosol). Additionally, the reanalyses adjust DOD and dust mass loading via data assimilation, but this will not be fed through to changes in dust emission, which remain an unconstrained model variable in the reanalyses. This means that despite the assimilation of satellite AOD retrievals, dust processes in CAMS and MERRA2 remain model-dependent and entail some level of uncertainty (Xian et al., 2020). Therefore, the comparisons between models and the reanalyses presented here should be interpreted with some caution.

## 2.4 Multiple linear regression

To investigate factors that drive dust emissions, we performed a multiple linear regression (ordinary least square) on standardised dust emissions and their drivers following Pu and Ginoux (2018). We started the regression with five dust emission drivers: bare soil fraction, leaf area index, precipitation, soil moisture, and surface winds. However, we deleted leaf area index and soil moisture from the regres-

sion after the variance inflation factor analysis, which indicates these two variables bear similar information to others included in the regression (not shown). We regressed grid cell dust emission to bare soil fraction, precipitation, and surface winds using standardised monthly mean data over the period 1985–2014 for each individual model and over 2005–2014 for CAMS. Note that due to data availability, we were only able to do this with 10 of the 16 CMIP6 models and CAMS (see Sect. 3.1.3).

We take the absolute value of the three regression coefficients as an indication of the importance of each in driving dust emission and measure their relative importance by normalising each coefficient against the sum of the three. Bootstrap resampling is used to test the significance of the regression coefficients. To demonstrate the relative importance of each driver at regional scales in different seasons, we repeat the regression for each individual calendar month using standardised regional and monthly mean data calculated from each of the 10 individual models (not shown) as well as the model ensemble mean. Note that here we use monthly data to feed the regression, while strong winds at shorter timescales may account for disproportionally more dust emissions. However, we were not able to test it due to the lack of high-resolution model outputs.

## 3  Results

### 3.1  Dust emission

We start with the climatology and global budget (Figs. 2 and 3 and Table 2) of dust emissions. We then show the seasonal cycles (Fig. 4) of dust emissions over the eight major source regions (see Fig. 1 for region definitions). Next, we look at the drivers of dust emissions in each individual model and CAMS reanalysis at the global scale (Fig. 5), and we demonstrate the relative importance of each driver in different seasons at regional scales (Fig. 6 for MEM and Fig. S2 in the Supplement for CAMS). We focus primarily on the differences between individual models, while comparing them to the CAMS and MERRA2 reanalyses. We also compare and discuss our results with previous-generation ESMs, with CMIP5 (Evan et al., 2014; C. Wu et al., 2020; M. Wu et al., 2020) and AeroCom III models (Gliß et al., 2021) in particular.

### 3.1.1  Climatology

Figure 2 shows the 2005–2014 global and annual mean climatology of dust emissions in each individual model; the MEM is shown in Fig. 3a and is compared to CAMS (Fig. 3b) and MERRA2 (Fig. 3c). Models generally capture the dust emission hotspots, namely the so-called "dust belt" (Ginoux et al., 2012) that extends from North Africa, the Middle East, Central Asia, and South Asia to East Asia. However, similar to CMIP5 models (C. Wu et al., 2020), there are considerable differences between models in other source regions such as North and South America and Australia. Models show significant differences in the intensity and spatial heterogeneity of dust emissions, reflecting the structural differences in dust emission schemes implemented in different models. Particularly, CESM2 models (Fig. 2a–d) and NorESM2-LM (Fig. 2o) put dust emissions in a few grid cells, whereas dust emissions in HadGEM-GC31-LL (Fig. 2i), INM-CM4-8 (Fig. 2j), and INM-CM5-0 (Fig. 2k) models have relatively homogeneous spatial patterns and significantly larger source areas (also see Fig. 3e). The spatial pattern and magnitudes of dust emission in the MEM are in broad agreement with CAMS and MERRA2. Pronounced differences are however found outside major desert dust source regions. This is because a few models (HadGEM-GC31-LL, INM-CM4-8, and INM-CM5-0) emit dust over these regions that are rarely deemed as potential dust sources (Ginoux et al., 2012).

The CMIP6 models estimate that between 1.4 Pg (INM-CM4-8) and 7.6 Pg (MIROC-ES2L) dust is emitted into the atmosphere annually, producing a MEM estimate of around 3.5 Pg yr$^{-1}$. The CMIP6 model range is just as large as the CMIP5 models (0.7–8.2 Pg yr$^{-1}$; C. Wu et al., 2020) and is much larger than the AeroCom phase III models (0.8–5.7 Pg yr$^{-1}$) and the recent observationally constrained estimates of $\sim 5$ Pg yr$^{-1}$ for PM$_{20}$ dust particles (Kok et al., 2021). Twelve of the 16 models simulate significantly more dust emissions than CAMS and MERRA2 reanalyses ($\sim 1.6$ Pg yr$^{-1}$; Fig. 3d), making the MEM more than 2 times larger than CAMS and MERRA2. Noticeably, in CMIP5, HadGEM2-CC has the most dust emission (8.2 Pg yr$^{-1}$), which is replaced by UKESM1-0-LL (7.5 Pg yr$^{-1}$) and MIROC-ES2L (7.6 Pg yr$^{-1}$) in CMIP6, while HadGEM-GC31-LL (3.3 Pg yr$^{-1}$) lies very close to the CMIP6 MEM estimate. The 2-fold difference in dust emissions between UKESM1-0-LL and HadGEM-GC31-LL is attributable to the additional Earth system interactions such as the dynamic vegetation included in UKESM1-0-LL (Mulcahy et al., 2020). This demonstrates the strong impact of model complexities on simulated dust emissions.

North Africa contributes the most (57 %, minimum–maximum: 28 %–69 %) to global dust emissions in CMIP6 models (Fig. 3d), generally agreeing with CAMS (46 %) and MERRA2 (60 %). This is followed by the Middle East (17 %, 11 %–25 %) and North China (9 %, 2 %–16 %). It is worth noting that the contribution of North Africa to global dust emission may be overestimated, while those of the Middle East and North China may be underestimated (Kok et al., 2021). These three regions make up more than 80 % of global dust emissions, while models disagree fundamentally about the relative contributions of other source regions. Particularly, South Asian dust emission is only important in UKESM1-0-LL, contrasting to the very limited dust emissions in all other models in this region including HadGEM-GC31-LL.

**Table 2.** Global total budgets for dust emission, deposition, and burden. Also shown are dust lifetime and global annual mean DOD.

| | Emission (Tg yr$^{-1}$) | Total deposition (Tg yr$^{-1}$) | | Dry deposition (Tg yr$^{-1}$) | | Wet deposition (Tg yr$^{-1}$) | | Burden (Tg) | Lifetime (days) | DOD (#) |
|---|---|---|---|---|---|---|---|---|---|---|
| | | Total | Land[a] | Total[b] | Land[c] | Total[d] | Land[e] | | | |
| CESM2 | 2238 | 2150 | 1606 (80) | 782 (36) | 659 (82) | 1369 (64) | 947 (69) | 27 | 4.6 | 0.027 |
| CESM2-FV2 | 2577 | 2133 | 1542 (72) | 768 (36) | 626 (82) | 1366 (64) | 917 (67) | 26 | 4.4 | 0.025 |
| CESM2-WACCM | 2210 | 2100 | 1578 (75) | 769 (37) | 649 (84) | 1341 (63) | 929 (69) | 27 | 4.7 | 0.026 |
| CESM2-WACCM-FV2 | 7050 | 5835 | 4553 (78) | 2202 (38) | 1794 (81) | 3633 (62) | 2351 (65) | 74 | 4.6 | 0.073 |
| CNRM-ESM2-1 | 2655 | 2424 | 1926 (79) | 1672 (68) | 1415 (85) | 753 (32) | 511 (68) | 14 | 2.1 | 0.011 |
| CanESM5 | 3274 | 2381 | 2082 (87) | 2056 (86) | 1830 (89) | 325 (14) | 252 (77) | 10 | 1.5 | 0.027 |
| GISS-E2-1-G | 1639 | 1586 | 1234 (78) | 1055 (67) | 924 (88) | 531 (33) | 311 (59) | 23 | 5.3 | 0.023 |
| GISS-E2-2-G | 1560 | 1510 | 1133 (75) | 1022 (68) | 880 (86) | 488 (32) | 253 (52) | 28 | 6.8 | 0.028 |
| HadGEM3-GC31-LL | 3255 | 3251 | 2912 (89) | 2700 (83) | 2591 (96) | 551 (17) | 321 (58) | 14 | 3.4 | 0.016 |
| INM-CM4-8 | 1374 | 1349 | 923 (68) | 851 (63) | 654 (77) | 498 (32) | 269 (54) | | | 0.033 |
| INM-CM5-0 | 1414 | 1385 | 936 (68) | 865 (62) | 660 (76) | 520 (32) | 276 (53) | | | 0.034 |
| IPSL-CM6A-LR | | | | | | | | 21 | | 0.033 |
| MIROC-ES2L | 7571 | 5852 | 5003 (85) | 4978 (85) | 4451 (89) | 902 (15) | 552 (61) | 31 | 1.9 | 0.045 |
| MRI-ESM2-0 | 5725 | 5473 | 4512 (82) | 3575 (65) | 3253 (91) | 1898 (35) | 1259 (66) | 27 | 1.8 | 0.027 |
| NorESM2-LM | 7092 | | | | | | | 9 | | 0.030 |
| UKESM1-0-LL | 7453 | 7443 | 6923 (93) | 6518 (88) | 6288 (96) | 925 (12) | 635 (69) | 18 | 0.87 | 0.011 |
| AMIP MEM | 3472 | 3201 | 2594 (81) | 2123 (66) | 1895 (89) | 1078 (34) | 699 (65) | 25 | 2.8 | 0.029 |
| CAMS | 1624 | 6598 | 5119 (78) | 4548 (69) | 4069 (89) | 2050 (31) | 1050 (51) | 12 | 0.66 | 0.019 |
| MERRA2 | 1594 | 1607 | 1155 (72) | 1123 (70) | 930 (83) | 485 (30) | 225 (46) | 23 | 5.2 | 0.030 |

[a] Numbers in brackets are percentages of total depositions to land relative to global total (dry + wet) depositions. [b] Numbers in brackets are percentages of total dry depositions relative to global total (dry + wet) depositions. [c] Numbers in brackets are percentages of total dry depositions to land relative to global total dry depositions. [d] Numbers in brackets are percentages of total wet depositions relative to global total (dry + wet) depositions. [e] Numbers in brackets are percentages of total wet depositions relative to global total wet depositions.

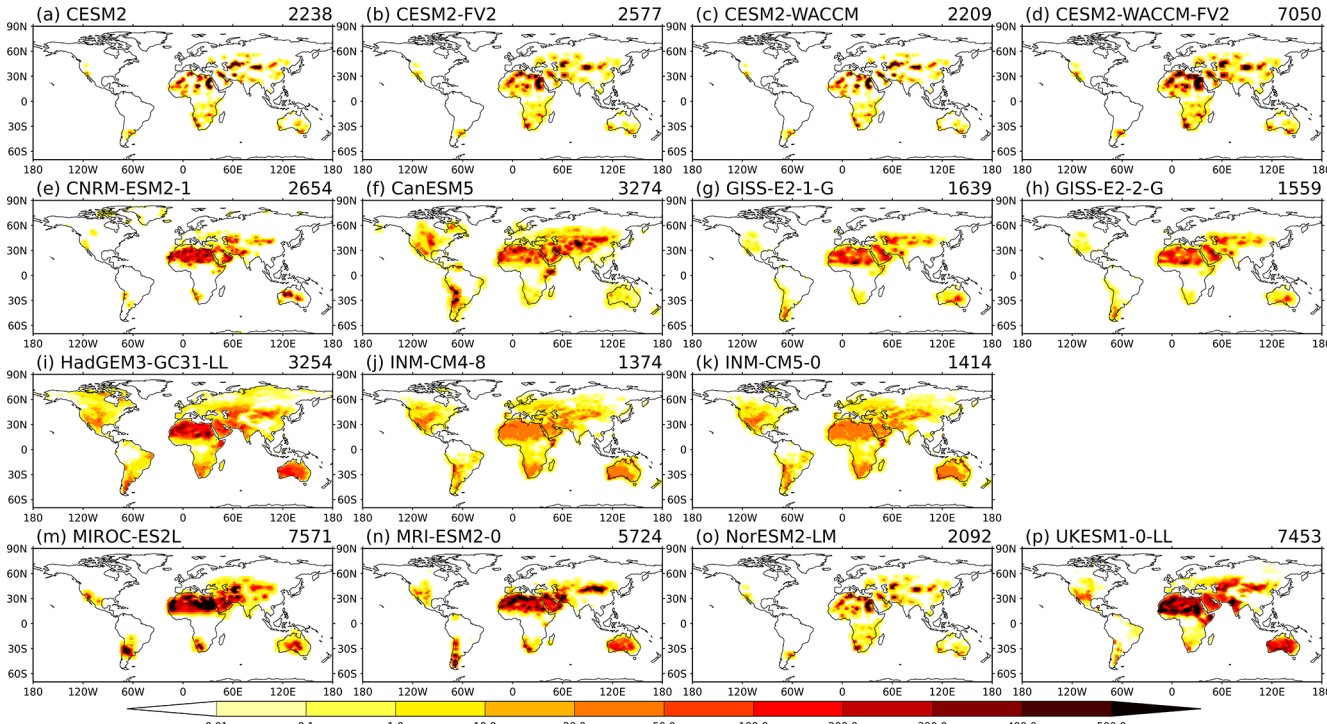

**Figure 2.** The CMIP6 AMIP models' simulated global annual mean (2005–2014) dust emission (g m$^{-2}$ s$^{-1}$) climatology. The numbers on the top right of each panel denote the global total dust emission budget (Tg yr$^{-1}$).

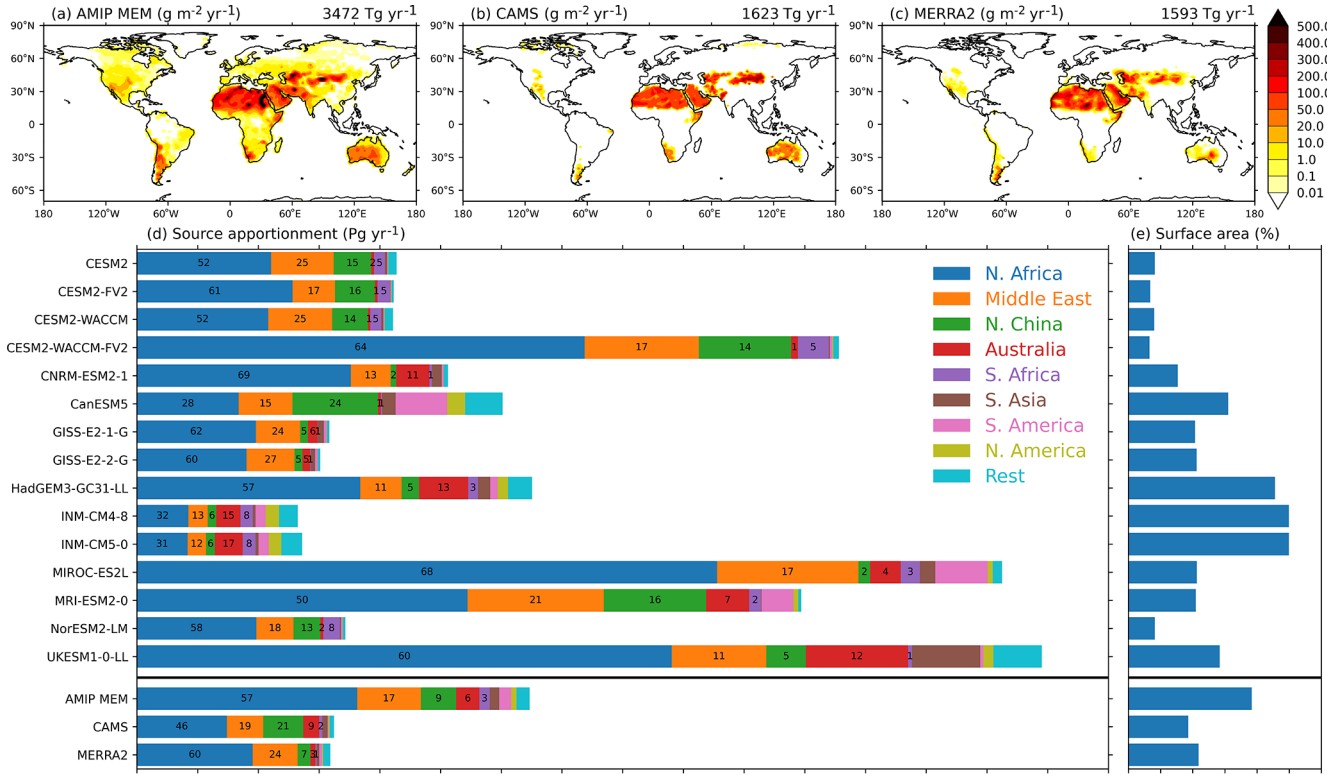

**Figure 3.** Intercomparison of 2005–2014 annual mean dust emissions between models and reanalyses. Maps show the annual mean dust emissions ($\mathrm{g\,m^{-2}\,yr^{-1}}$) from **(a)** AMIP MEM, **(b)** CAMS, and **(c)** MERRA2. The numbers on the top right of each panel denote the global total dust emission budget ($\mathrm{Tg\,yr^{-1}}$). The global annual total dust emission budgets (**d**; $\mathrm{Pg\,yr^{-1}}$) and the fraction of total dust emission areas relative to the global surface area (**e**; %) are shown for each individual model as well as for the AMIP MEM, CAMS, and MERRA2. The contributions of major dust source regions are coloured out in panel **(d)**, where the percentage contributions of the first five largest source regions are given at the centres of each coloured bars.

CMIP6 models also feature diversities in the global surface area of dust emissions (Fig. 3e), with the smallest area (around 2.5 % of the global surface area) found in the CESM2 family models and NorESM2-LM and the largest area found in INM-CM4-8 (15.0 %). The MEM estimate (11.5 %) is almost 2 times larger than that of MERRA2 (6.6 %) and CAMS (5.6 %). The range of the CMIP6 model estimates (2.5 %–15.0 %) is almost as large as that of the CMIP5 models (2.9 %–19 %) as reported by C. Wu et al. (2020). Note that the models that have the highest dust emissions do not necessarily have the largest emission areas, and vice versa. This again suggests the large diversities in dust emission intensities in different models.

### 3.1.2 Seasonal cycles

Figure 4 shows the normalised seasonal cycles of dust emissions over the eight source regions (see Fig. 1 for definitions of these regions); the absolute seasonal cycle profiles can be found in Fig. S1. The MEM agrees well with CAMS and MERRA2 in reproducing the patterns of the seasonal cycles. However, noticeable discrepancies are found between

individual models and reanalysis over a few key regions including North China (Fig. 4d), North America (Fig. 4g), and South America (Fig. 4h). Dust emission peak seasons in these regions have the large diversities between models. Two models behave very differently from others. First, the CanESM5 model simulates very different dust emission peak seasons over the Middle East (Fig. 4c), North China, South Asia, and North America. Second, MIROC-ES2L is the only model that simulates a summer peak, while all others show a spring peak over North Africa (Fig. 4a).

The seasonal cycles feature a double peak over North China and North America in a few models (CanESM5, MIROC-ES2L, INM-CM4-8, INM-CM5-0, MRI-ESM2-0), while the reanalysis datasets and most other models and reanalysis datasets present a single spring emission peak. The underlying mechanisms that drive these double peaks differ between these two regions. The models that simulate a secondary autumn emission peak in North China are found to have too strong surface winds in autumn and winter. In comparison, a deficit in autumn precipitation explains the secondary emission peak in North America (figures not shown).

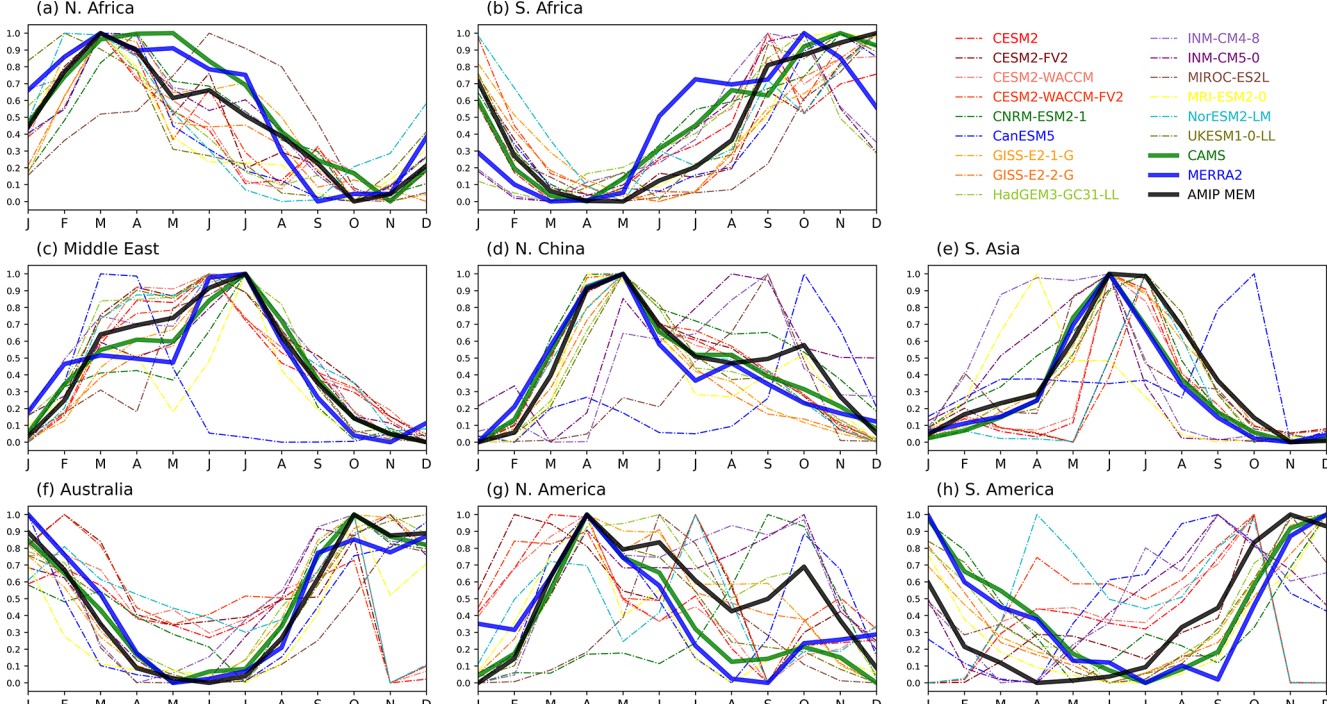

**Figure 4.** Normalised seasonal cycles of dust emissions over the eight dust source regions. Dashed curves represent individual models, while the AMIP MEM is shown in solid black. Also shown are results from CAMS (solid green) and MERRA2 (solid blue). The absolute dust emission seasonal cycles are included in Fig. S1.

The magnitude of the seasonal cycles in MEM and most individual models is much larger (up to 10 times) than those in CAMS and MERRA2 (Fig. S1) which lie very close to the lower bounds of the AMIP multi-model spreads. This is consistent with our finding above that the CMIP6 models have considerably more dust emissions than reanalysis (Sect. 3.3.1). Particularly, UKESM1-0-LL may overestimate dust emissions in South Asia and Australia (Fig. S1e, f). By contrast, the INM-CM4-8 and INM-CM5-0 models may have too little dust emission over North Africa (Fig. S1a).

In short, the seasonal cycles of dust emissions over major source regions are well reproduced by the MEM and most individual models, but a few models behave very differently to others over North China, North America, and South America.

### 3.1.3 Drivers

Figure 5 shows the dominant driver of dust emissions at each grid cell in the 10 CMIP6 models and the CAMS reanalysis, based on the methodology set out in Sect. 2.4. Surface wind speed is shown as the dominant driver of dust emissions in all the models and CAMS. This is consistent with previous studies (Evan, 2018; Pu and Ginoux, 2018). Precipitation only dominates dust emissions in the INM-CM4-8 and INM-CM5-0 models over a few regions (Fig. 5i, j). Our finding is consistent with Pu and Ginoux (2018), who show

that surface winds and precipitation are two of the most important factors determining seasonal DOD variations in the CMIP5 models.

We further examine the relative importance of each individual driver and their seasonal variations over the eight major dust source regions in the MEM (Fig. 6). The dominant role of surface winds in driving dust emission can be seen over North Africa (Fig. 6a), the Middle East (Fig. 6c), and North China (Fig. 6d) throughout the year, whilst surface bareness also plays an important role in other regions. Precipitation (influencing soil moisture) is shown to influence dust emission in North Africa, with the largest impact found in summer. Similarly, precipitation plays an important role in South Asian dust emission around the post-monsoon season in October (Fig. 6e). In other regions, precipitation shows a relatively minor contribution. Similar conclusions can be drawn for CAMS as a comparison (Fig. S2).

Overall, we found that surface wind and bareness are the first two most important factors in driving dust emissions in CMIP6 models and CAMS. We hence speculate that future changes to dust emissions may be most sensitive to changes in the surface wind related to circulation changes in combination with vegetation changes, rather than the frequency or severity of droughts, though the latter may indirectly influence future dust emissions via vegetation changes. However, it should be noted that surface bareness was found to play the most important role in controlling DOD over many dusty

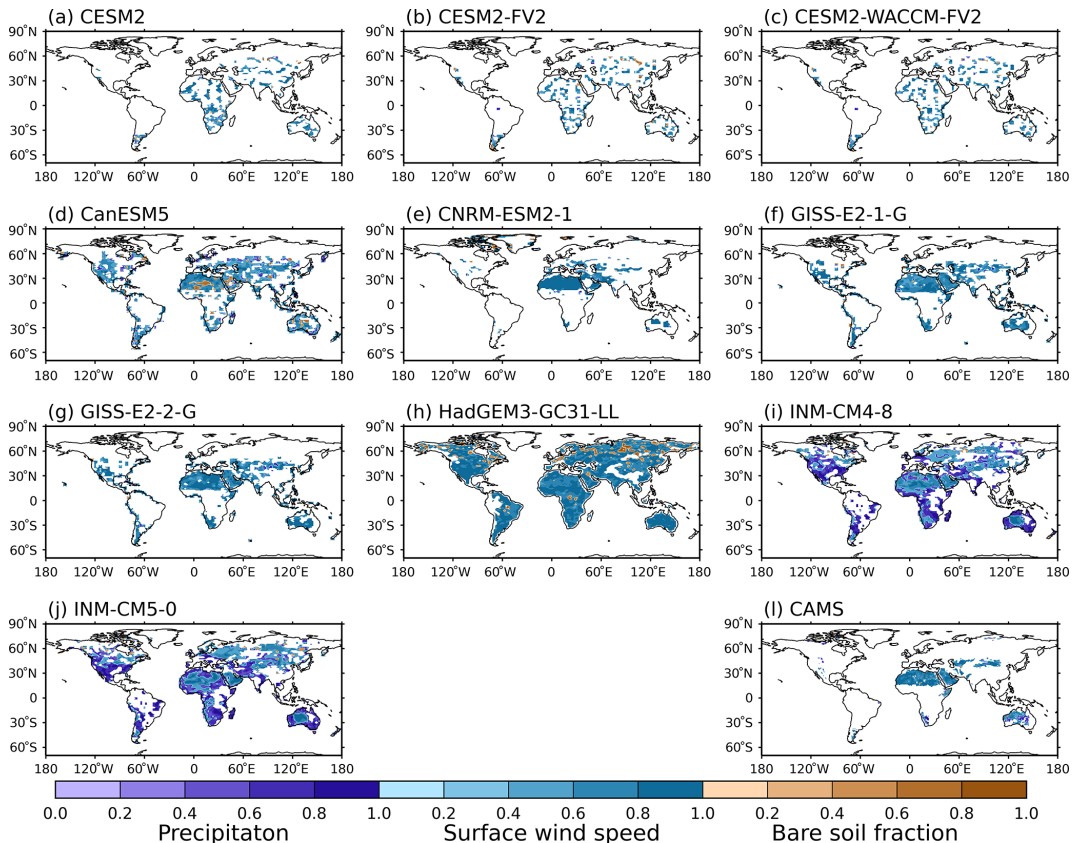

**Figure 5.** The dominant driver of dust emission and its relative importance (scaled to 0–1) in **(a–k)** models and **(l)** CAMS reanalysis. Purple for precipitation, blue for surface wind speed, and brown for bare soil fraction. Data used for regressions are 30 years (1985–2014) for models and 10 years (2005–2014) for CAMS due to data availability. Surface wind speed and precipitation in panel **(l)** are taken from ERA5.

regions in observations (Pu and Ginoux, 2018). The fact that models do not simulate significant trends in surface bareness over time explains why surface bareness plays a secondary role compared to surface winds in driving dust emissions in models, since the required criteria for dust emission are already satisfied.

## 3.2   Dust loading, deposition, and lifetime

This section examines atmospheric dust mass loading (Figs. 7, S3, S4) and deposition fluxes (Figs. 8, 9, and S5–S8) which in combination determine the atmospheric dust lifetime (Fig. 10). The global budgets of these fluxes in each model are summarised in Table 2.

The spatial pattern of the dust mass loading climatology in MEM (Fig. 7a) shows good agreement with CAMS (Fig. 7b) and MERRA2 (Fig. 7c), while the magnitudes are slightly larger. The global total atmospheric dust burden in MEM (25 Tg) is comparable to that of MERRA2 (23 Tg) and is 2 times larger than that of CAMS (12 Tg). Most models' simulated global total dust burden lies well around the MEM and reanalysis estimates (Fig. 7d). However, the CESM2-WACCM-FV2 model simulates significantly more dust in the

atmosphere (74 Tg). The range of global total dust burden in CMIP6 with the CESM2-WACCM-FV2 model excluded is 9–28 Tg. This is larger than that of the AeroCom III models (6–22 Tg; Gliß et al., 2021) but smaller than the CMIP5 models (3–42 Tg; C. Wu et al., 2020).

We calculated the meridional mean profiles of dust mass loading and DOD (Fig. 7e, f) to examine the gradients associated with dust transport from land to the adjacent oceans from the two largest source regions: North Africa and North China (boxes in Fig. 7c). Note here that we show the DOD profiles because more model and observational data are available for comparison, but similar conclusions can be drawn if one looks at the mass loading profiles (Fig. S4). Most models, and particularly the MEM, reproduce well the gradients compared to both reanalysis and satellite observations. However, the magnitudes of the profiles are too low in INM-CM5-0 and UKESM1-0-LL but too high in CESM2-WACCM-FV2 over the Africa–Atlantic region (Fig. 7e). By contrast, many models (CNRM-ESM2-1, HadGEM3-GC31-LL, MIROC-ES2L, MRI-ESM2-0, INM-CM4-8, INM-CM5-0, and UKESM1-0-LL) do not produce ample gradients over the Asia–Pacific region (Fig. 7f).

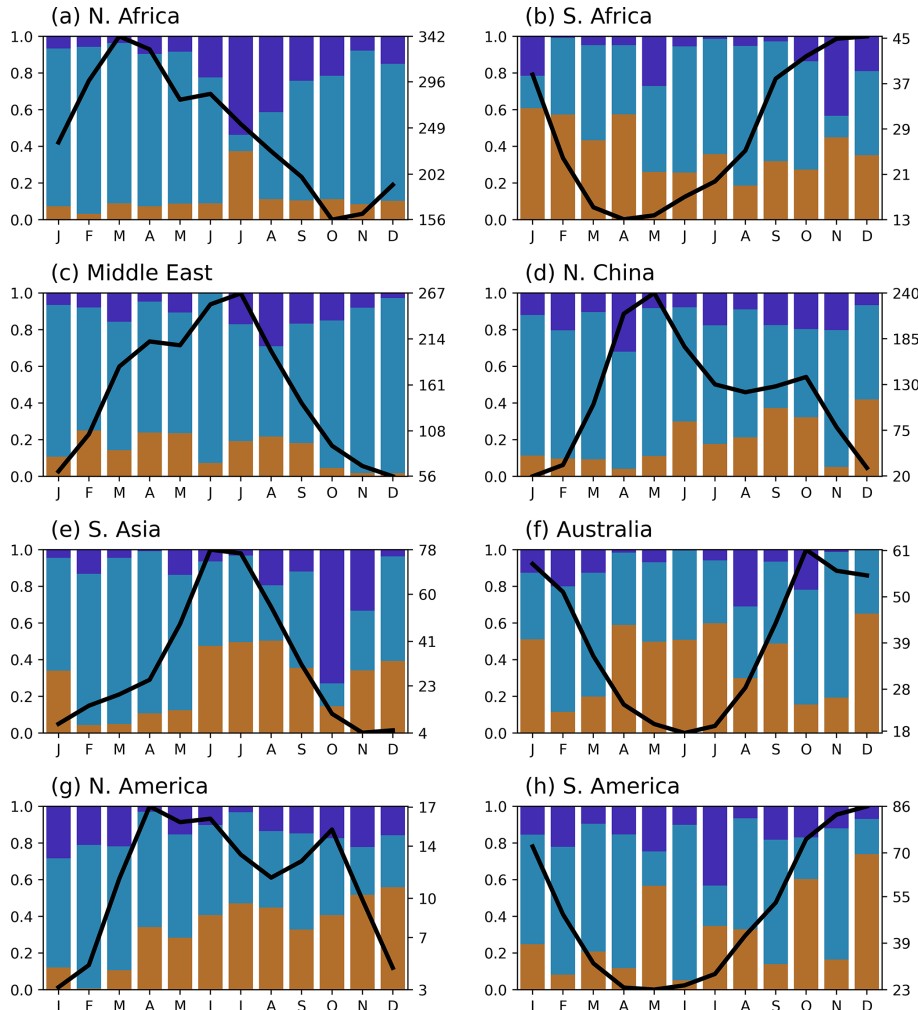

**Figure 6.** Normalised relative importance (left axis) of the three major dust emission drivers throughout the year over the eight major source regions in the MEM. Purple for precipitation, blue for surface wind speed, and brown for bare soil fraction. The black curves are AMIP MEM (models in Fig. 5) seasonal cycles of dust emissions (right axis; $mg\,m^{-2}\,d^{-1}$).

Figure 8 shows the climatology of the total (dry + wet) dust deposition flux and the percentage of wet deposition. Globally, CMIP6 models estimate that $\sim 3.5$ (1.3–7.4) Pg dust is removed from the atmosphere annually, which is consistent with the MERRA2 (1.6 Pg yr$^{-1}$) and CAMS (6.6 Pg yr$^{-1}$) estimates. As summarised in Table 2, dust is predominantly removed by dry deposition (60 %–86 %) in most models, agreeing with CAMS (69 %) and MERRA2 (70 %), yet the CESM2 models show that most ($\sim 74$ %TS3) of the total dust removal is via wet processes (also see Fig. S7). Dust removal over the oceans is controlled by wet deposition, while dry processes dominate over lands (Fig. 8b, d, f). The intercomparisons between reanalyses, models, and ground observations of total dust deposition fluxes (Fig. 9) show that CAMS and MERRA2 give fair representations of dust deposition compared to the observations (i.e. with a log-space root mean square error (RMSE) of $\sim 2.0$). Meanwhile, the

MEM and most individual models (Fig. S8) are as good as reanalyses. We note however the observational dust deposition fluxes only include PM$_{10}$ particles, while models and reanalyses datasets have larger dust particles. Therefore, whilst biases are expected, we are not able to quantify them due to the lack of size-resolved dust deposition fluxes from models in the CMIP6 archive.

Figure 10 summarises the differences in global dust burden, deposition fluxes, and lifetime (global dust burden divided by total deposition fluxes) between models and analysis datasets. Dust lifetime in CMIP6 models ranges by a factor of 4 from around 1.8 to 6.8 d, with a MEM estimate of 4.3 d. The range is larger than the CMIP5 models (1.3–4.4 d; C. Wu et al., 2020) and is just as large as the AeroCom III models (1.4–7.0 d; Gliß et al., 2021). MERRA2 (5.2 d) lies within the range of the CMIP6 model estimates. In comparison, CAMS is likely to underestimate dust life-

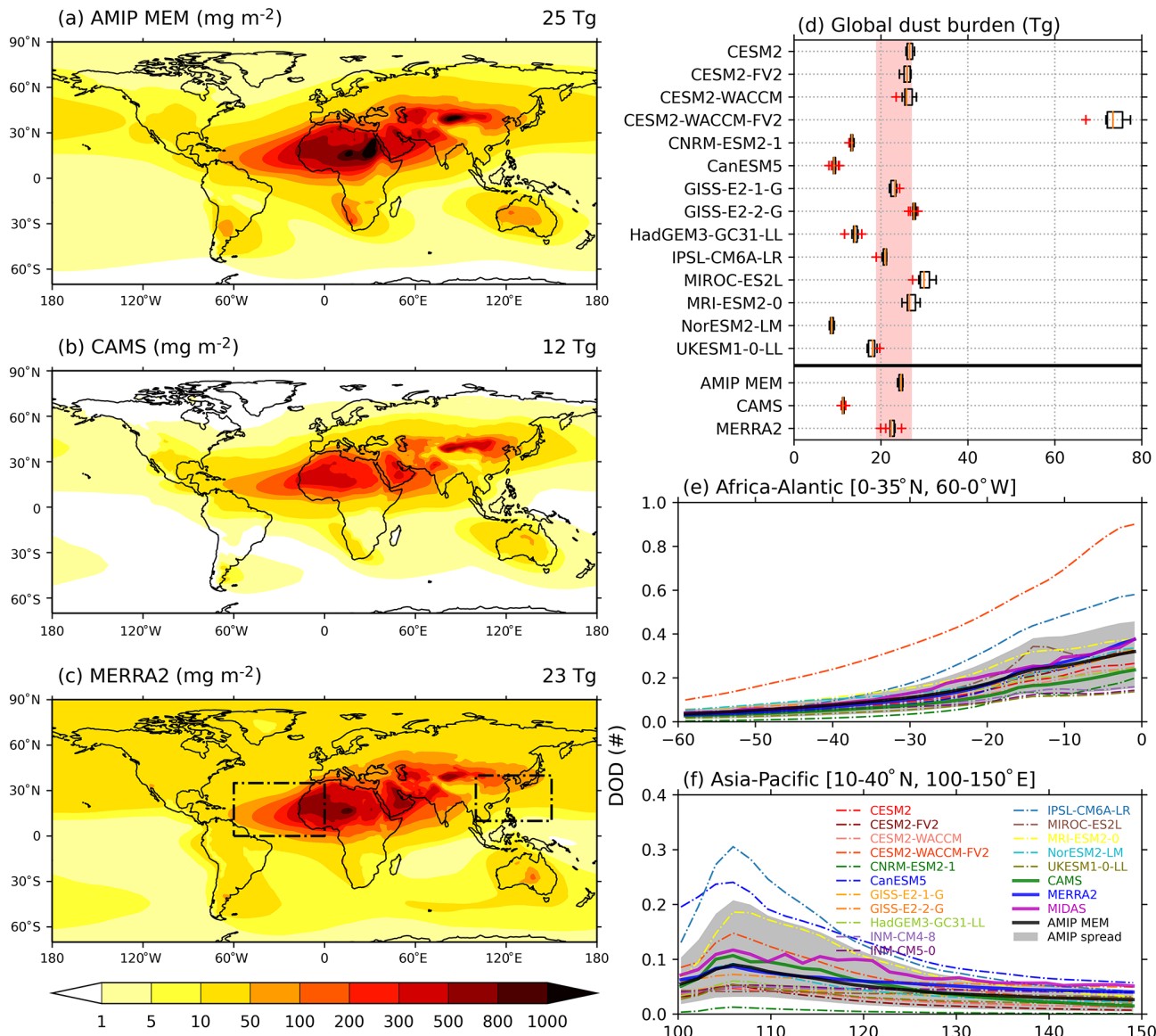

**Figure 7.** Intercomparison of 2005–2014 annual mean dust mass loading (mg m$^{-2}$) between **(a)** AMIP MEM, **(b)** CAMS, and **(c)** MERRA2. The numbers on the top right of each panel denote the global total dust burden (Tg). Maps for individual models can be found in Fig. S3. **(d)** Global total dust burden from each individual model as well as those of panels **(a)**–**(c)**: boxes denote the 10th–90th percentiles of the annual variability; red pluses denote outliers that are outside 1.5 times the annual standard deviation. The vertical pink shading represents the 10th–90th percentiles of the multimodal spread. Also shown is the meridionally averaged DOD over **(e)** the Africa–Atlantic region (0–35° N, 60–0° W; box in panel **c**) in June–July–August and **(f)** the Asia–Pacific region (10–40° N, 100–150° E; box in panel **c**) in April–May–June.

time (0.6 d) because of high deposition rates which lead to less dust in the atmosphere (see above). We found a linear relationship between dust lifetime and the ratio of global dry-to-total depositions across different models (Fig. 10b). This suggests that dry processes control dust lifetime, which is further demonstrated by the strong (weak) linear correlation between dust lifetime and dry (wet) deposition across different datasets (Fig. 10c, d). That is, the fewer the dry processes, the longer dust resides in the atmosphere before being finally removed by the relatively infrequent wet depositions events,

occurring mostly further from dust sources compared to dry deposition.

## 3.3   AOD and DOD

In this section, we turn to DOD that is associated with the above-presented dust processes. In addition to evaluating CMIP6 model-simulated DOD, we also examine whether models' performance in simulating the optical depth of dust differs from those of other aerosol species. We first show

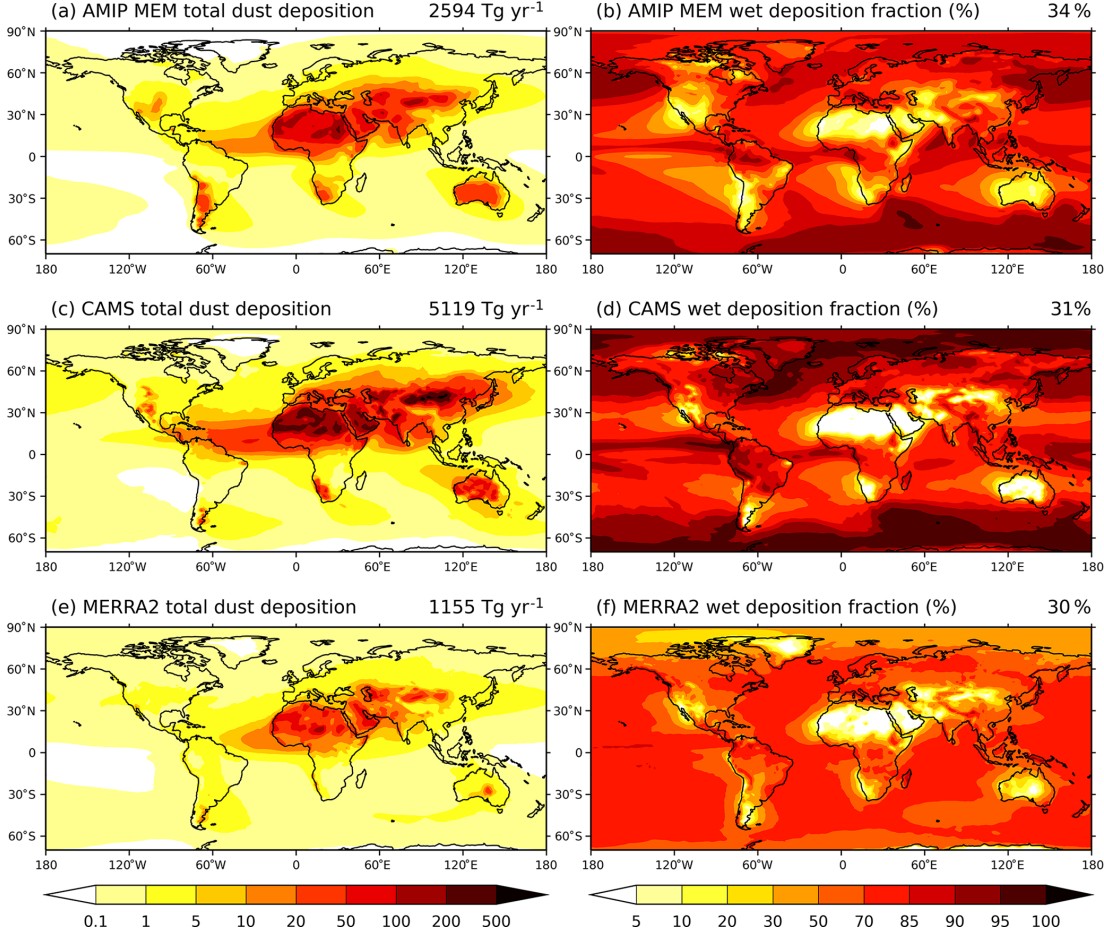

**Figure 8.** Intercomparison of 2005–2014 mean of annual total (dry + wet) dust deposition (**a, c, e**; $g\,m^{-2}\,yr^{-1}$) and the ratio of wet-to-total depositions (**b, d, f**; %). (**a, b**) AMIP MEM, (**c, d**) CAMS, and (**e, f**) MERRA2. The numbers on the top right of each panel denote the global total dust deposition flux to land ($Tg\,yr^{-1}$) and the fraction of global wet-to-total dust depositions (%).

differences in model-simulated AOD (Fig. 11) and DOD (Figs. 12–13) and then examine where such differences come from (Fig. 14). Finally, we examine the seasonal cycles of DOD, whilst comparing them with the seasonal cycles of dust emissions, over the eight major dust source regions (Fig. 15).

Figure 11 shows an intercomparison of the AOD climatology between the MEM and those from satellite and reanalyses. The AOD climatology in each individual model is included in Fig. S9. The MEM, and most individual models, reproduces the AOD climatology well. However, a few models (GISS-E2-1-G, GISS-E2-2-G, INM-CM4-8, INM-CM5-0) struggle to capture the AOD spatial pattern: the spatial correlation ($R^2$) between these models and satellite and reanalysis datasets are less than 0.2 (Fig. 11h). By contrast, the HadGEM3-GC31-LL, MIROC-ES2L, MRI-ESM2-0, and UKESM1-0-LL models' simulated AOD have even greater spatial correlations to satellite and reanalysis datasets compared to the MEM. Compared to AOD, the MEM DOD climatology (Fig. 13a) has slightly greater spatial corre-

tions with satellite ($R^2 = 0.76$) and reanalyses ($R^2 = 0.85$ and 0.86). However, the models that simulate the spatial pattern of DOD well do not necessarily perform well in simulating AOD (comparing Fig. 11h to Fig. 13h). For example, the two GISS models (denoted by letters g and h, also see Fig. S9g, h) are top-ranked in capturing the spatial pattern of DOD (Fig. 13h) but are lowest-ranked in simulating the spatial pattern of AOD (Fig. 11h). This highlights the inconsistent behaviour of CMIP6 models in simulating the optical depth of different aerosol species. However, it may also question the reliability of the MIDAS retrievals which are largely based upon MERRA2. More importantly, careful comparisons between Figs. 11e–g and 13e–g indicate that the magnitudes of DOD have larger biases than AOD.

CMIP6 model-simulated global mean DOD varies by a factor of 7 from 0.011 to 0.073, with the MEM estimate of 0.029. This is consistent with an observationally constrained estimate of $0.030 \pm 0.005$ for $PM_{20}$ dust (Ridley et al., 2016), 0.033 (0.031–0.040) in MIDAS, and 0.031 (0.028–0.036) in MERRA2 but is $\sim 1.5$ times that of CAMS ($\sim 0.019$).

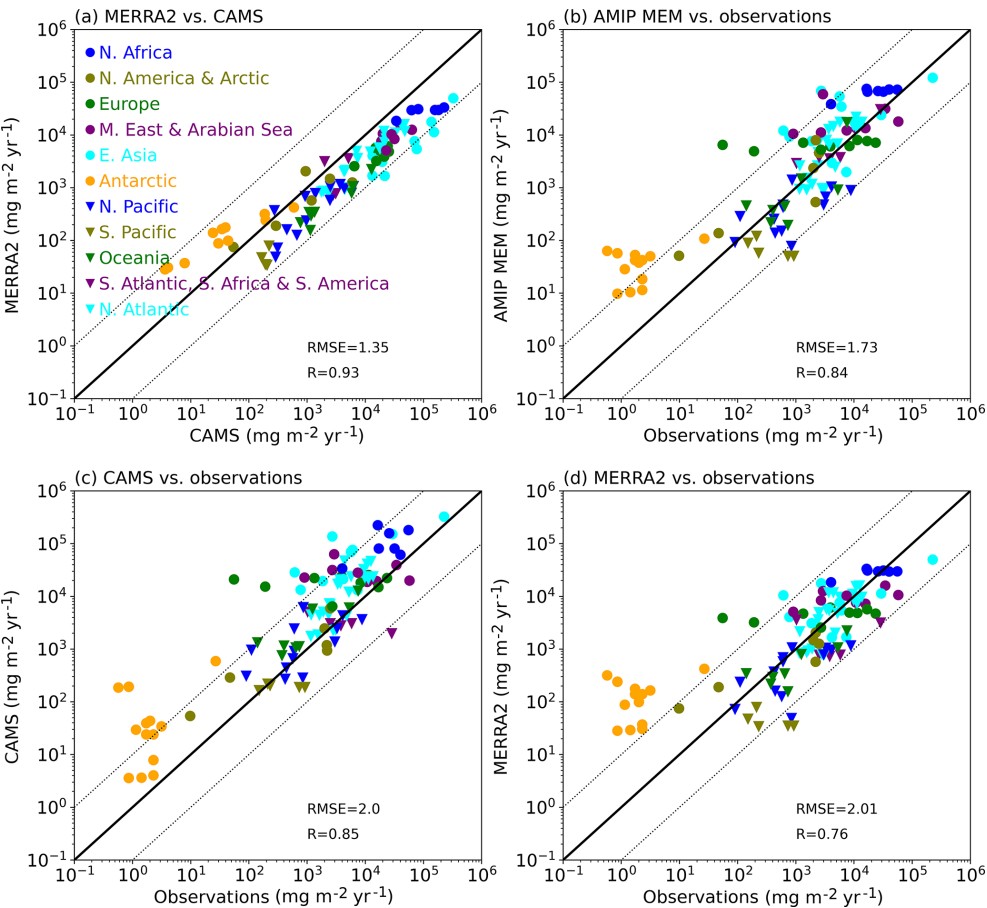

**Figure 9.** Scatterplots of annual mean total dust deposition flux at ground stations between **(a)** MEERA2 and CAMS, **(b)** AMIP MEM and observations, **(c)** CAMS and observations, and **(d)** MERRA2 and observations. The stations are marked with different styles and colours for different locations (see Fig. 1). The correlation coefficients and root mean square errors (RMSE) are calculated in log space. The 1 : 1 (solid) and 1 : 10/10 : 1 (dotted) lines are plotted for reference. The scatterplots between each individual model and the observations can be found in Fig. S8.

There are however significant biases in the MEM-simulated DOD magnitudes at regional scales. For example, models tend to overestimate DOD, which can be seen over the Sahara and the Chinese deserts in the MEM and in most models (Fig. 12). More specifically, the AMIP MEM estimate of the regional mean DOD in North Africa (0.278) is 1.2–1.7 times larger than those of satellite (0.228) and reanalysis datasets (0.165 in CAMS and 0.238 in MERRA2). Similarly, the MEM overestimates North China mean DOD (0.142) by 1.2–1.5 times compared to satellite (0.097) and reanalyses (0.118 in CAMS and 0.099 in MERRA2). Such overestimates can reach up to 4–5 times in a few models such as CESM2-WACCM-FV2 (Fig. 12d) and MIROC-ES2L (Fig. 12m). We note however that previous studies using CMIP5 and Aero-Com III models concluded that DOD in these regions were well captured (Pu and Ginoux, 2018; Gliß et al., 2021). Finally, it is important to point out that none of the 16 models, nor the reanalyses, are able to capture the dust transportation and atmospheric accumulation to the south of the Himalayas

over the Indo-Gangetic Plain as shown by satellite data (box in Fig. 13a). Also, none of the AMIP models captures the regional DOD variability over the Middle East, Central Asia, the Chinese desert, and eastern China (Fig. 13a).

We showed above the biases in model-simulated DOD magnitudes. Here we further investigate such biases by looking at the DOD probability density distributions of the models in comparison to satellite and reanalyses, as a function of the difference in the magnitude of the DOD. This gives an insight into how well the models represent weaker vs. heavier dust events. Model grid cell DOD values are grouped into three categories ($\leq 0.1$, 0.1–0.4, and $> 0.4$) to calculate the probability density function independently (Fig. 14a–c). Almost all models and the MEM underestimate small ($\leq 0.1$) DOD values (Fig. 14a) while significantly overestimating large ($> 0.4$) DOD values (Fig. 14c). By contrast, the moderate DOD values (Fig. 14b) are relatively well reproduced. We note that models overestimate small DOD values compared to CAMS, while the CAMS DOD is low compared to

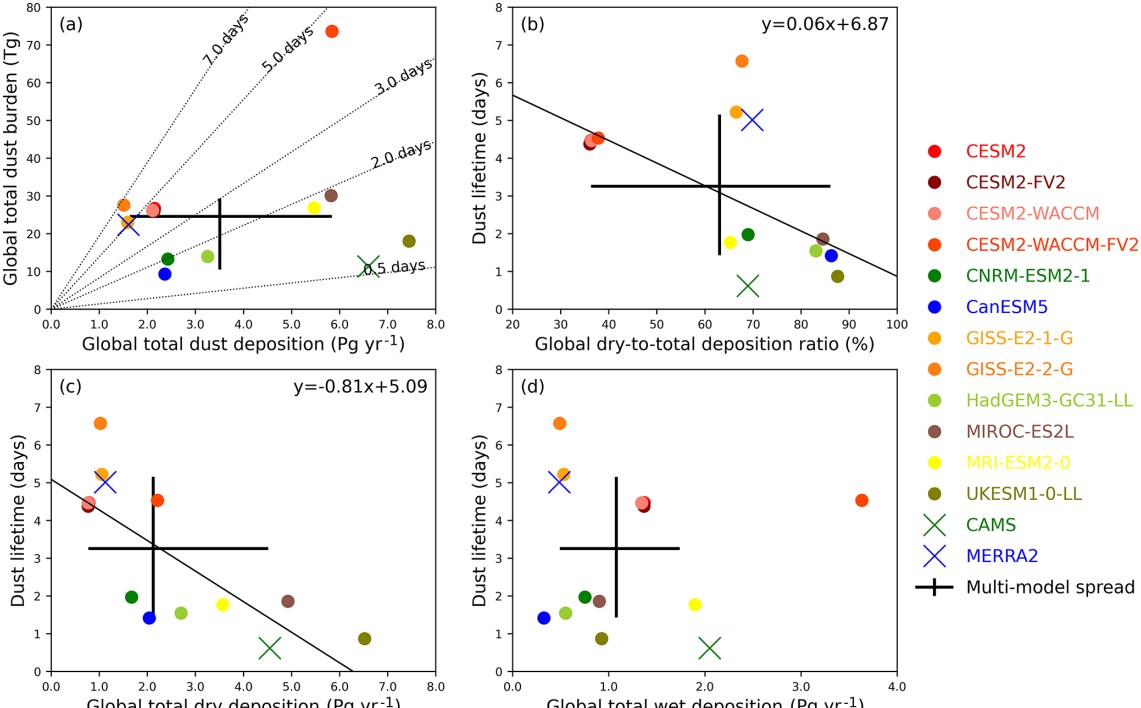

**Figure 10.** Scatterplots of **(a)** global annual mean total dust burdens (Tg) vs. annual total dust deposition (Pg yr$^{-1}$), and global dust lifetime (days) vs. **(b)** the ratio of global dry-to-total deposition (%), **(c)** total dry depositions (Pg yr$^{-1}$), and **(d)** total wet depositions (Pg yr$^{-1}$). Model colour codes are the same as in other figures, along with CAMS (green cross) and MERRA2 (blue cross). The AMIP multi-model mean and spread (10th–90th percentiles) are shown by the black pluses. The dotted slope lines in panel **(a)** denote dust lifetime intervals (days). The solid slope lines in panels **(b)** and **(c)** are the linear fitting between $x$ and $y$ axis using all data points. All results shown are 2005–2014 annual mean.

the other datasets (see above). To further understand where the biases in models' simulated DOD come from, we examine the global zonal mean DOD profile and the distribution of regional mean DOD over the eight major dust source regions in Fig. 14d. The overestimates (black crosses, denoting the MEM), compared to both satellite and reanalysis datasets (blue/green/purple crosses), can be seen over North Africa, North China, South Africa, and Australia. Meanwhile, DOD is underestimated over South Asia, which again implies that models fail to capture the dust accumulations in the Indian subcontinent.

The model-simulated seasonal cycles of DOD (Figs. 15 and S10) are in broad agreement with reanalysis and satellite observations. However, noticeable discrepancies are again found in South Africa (Fig. 15b), North China (Fig. 15d), North America (Fig. 15g), and South America (Fig. 15h) where there are also large inter-model discrepancies in model-simulated seasonal cycles of dust emission. By contrast, the Middle East (Fig. 15c) and South Asia (Fig. 15e) show good agreement between observed and model-simulated DOD seasonal cycles. Finally, it is interesting to note that the seasonal cycle of DOD over North Africa (Fig. 15a) peaks slightly later than dust emission. This may indicate the importance of dust transport in influencing dust

optical depth and its seasonal cycles. In comparison, the seasonal cycles of DOD are synchronised with dust emissions in MEM over all other regions. Therefore, model biases in dust emissions are likely to be reflected in DOD. In North Africa, there seems to be a split into two groups of models between the ones that sustain DOD over the summer versus the ones which drop off in June/July (Fig. 15a). Overall, similar to dust emissions, the seasonal cycles of DOD are well reproduced by most models, and the MEM does a good job in capturing DOD seasonal cycles in most regions. However, a few models still struggle to capture the seasonal cycles over North China, North America, and South America.

## 4 Conclusions and discussions

In this study, we examine dust aerosols in 16 state-of-the-art Earth system models participating in the CMIP6 AMIP. We evaluated models' present-day (2005–2014) dust aerosol processes (emission, deposition, burden, lifetime), as well as dust aerosol optical depth (DOD), against several global reanalysis and observational datasets. We presented our findings in the context of CMIP5 and AeroCom III models. Our key findings are the following.

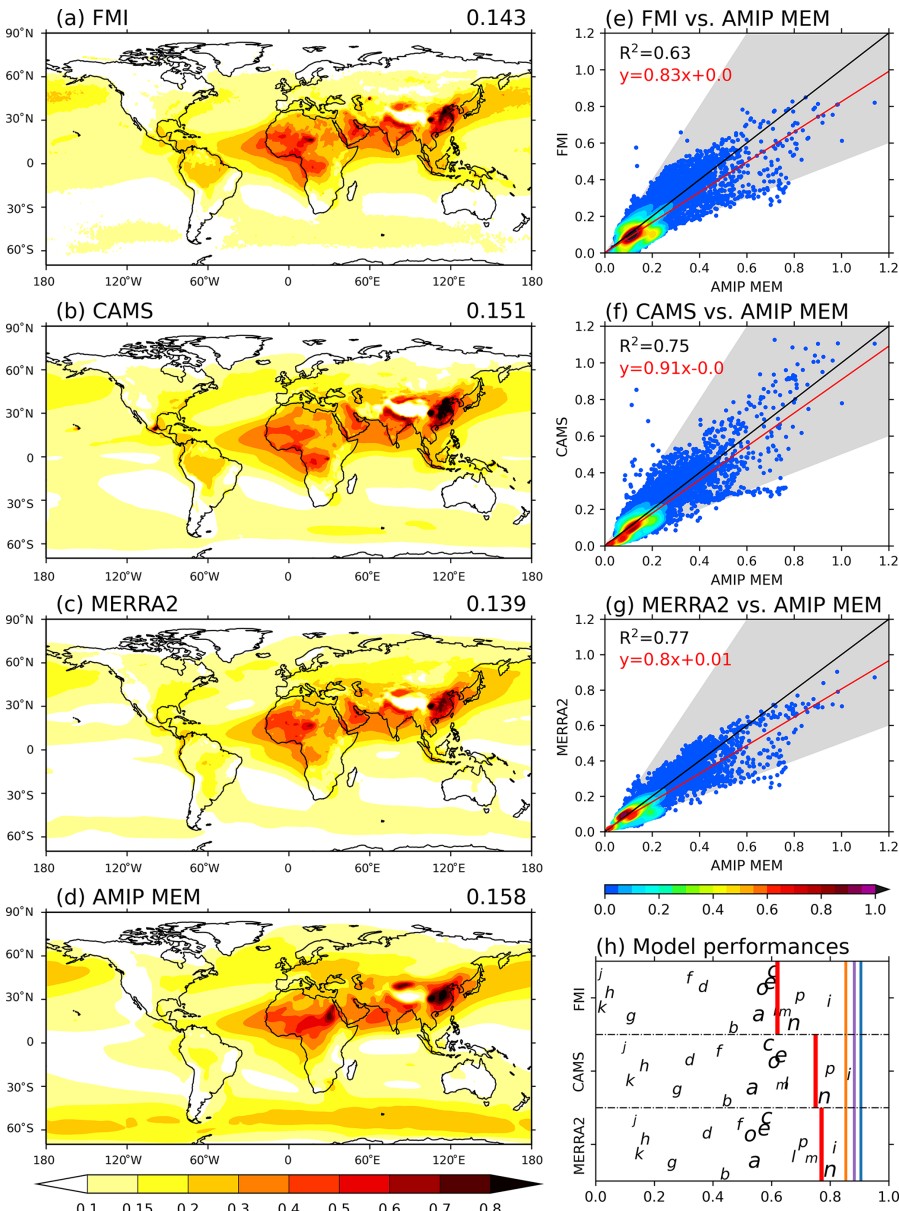

**Figure 11.** Intercomparison of 2005–2014 annual mean AOD from **(a)** FMI merged satellite retrievals, **(b)** CAMS, **(c)** MERRA2, and **(d)** AMIP MEM. Panels **(e)**–**(g)** show the density ($10^{-4}$) scatterplots of observation/reanalyses (**a–c**; $y$ axes) vs. AMIP MEM (**d**; $x$ axes): the black lines are the 1 : 1 correspondence while the red lines are the linear fitting ($R^2$ and the regression equation given at top left corners). Panel **(h)** is a summary of the performance of each individual model measured by the spatial correlation ($R^2$, $x$ axis) between each individual model and observation/reanalyses. The $y$ axes do not have any physical meaning and are used to make the plot readable. Models are shown by letters (see Table 1). The red vertical bars denote where the AMIP MEM stands. The blue vertical bar shows the spatial $R^2$ between CAMS and FMI. Similarly, orange bars are for MERRA2 vs. FMI, and purple bars are for CAMS vs. MERRA2.

– The CMIP6 models generally capture the spatial patterns of global dust emission, mass loading, and removal processes. However, large uncertainties and intermodel diversities (a factor of 4–5) are found in all these fields.

– The global dust emission is predominantly driven by surface winds (as opposed to bare soil fraction and precipitation) in models and the CAMS reanalysis.

– Most models, and particularly the MEM, capture dust seasonal cycles over major source regions. However, the seasonal cycles are poorly constrained over North China, North America, and South America.

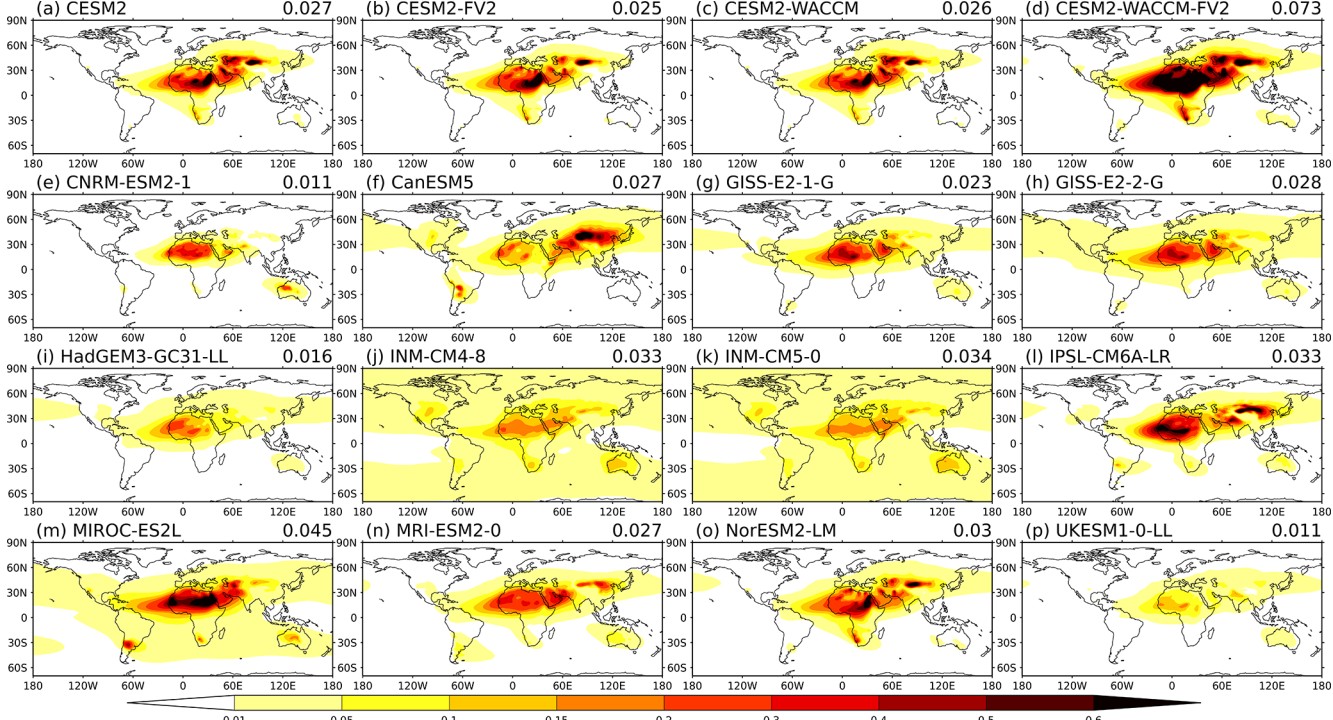

**Figure 12.** The CMIP6 AMIP models' simulated global annual mean (2005–2014) DOD climatology. The numbers on the top right of each panel denote the global means.

- CMIP6 models simulate large diversities in global mean DOD that range by a factor of 7. The MEM estimate however is consistent with both satellite and reanalyses.

- Almost all CMIP6 models underestimate small DOD values but significantly overestimate large DOD values compared to satellite and reanalyses; the overestimates are found mainly over the dustiest regions such as North Africa (1.2–1.7 times in MEM) and North China (1.2–1.5 times in MEM).

- The CMIP6 models consistently fail to capture certain key features of regional dust distributions – for example, atmospheric dust accumulation to the south of the Himalayas over the Indo-Gangetic Plain and regional DOD variability over East and Central Asia and the Middle East.

- Dust processes in CAMS and MERRA2 datasets are very uncertain, as demonstrated by the 2-fold difference in global dust load (23 Tg in MERRA2 vs. 12 Tg for CAMS).

Additionally, it is worth summarising the following quantitative findings.

- Around 3.5 (1.4–7.6) Pg of dust is emitted annually in the CMIP6 models, and the MEM estimate is double the amount in the MERRA2 and CAMS reanalyses. A

similar overestimate was highlighted in previous studies for CMIP5. Also, there are large diversities in the extent (2.5 %–15.0 % of global surfaces) and intensity of dust emission between models.

- North Africa, the Middle East, and North China combined make up over 80 % of global total dust emissions. However, models disagree considerably about the contributions from other dust source regions.

- There are large uncertainties in the global total dust burden (9–74 Tg) in the atmosphere; the estimated range is much larger than that in CMIP5 (3–42 Tg) and Aero-Com III (6–22 Tg) models.

- The global dust removal ($\sim 3.5$ (1.3–7.4) Pg yr$^{-1}$) is dominated by dry deposition processes (60 %–86 %) which are found to control global dust lifetime that varies by a factor of 4 from 1.8 to 6.8 d in CMIP6 models.

Overall CMIP6 models generally reproduce global dust processes, and the MEM performs better than most individual models. Nevertheless, large model uncertainties and diversities still exist. This may be associated with increases in model complexities, as demonstrated by the difference between HadGEM-GC31-LL and UKESM1-0-LL (Mulcahy et al., 2020). Models still suffer from deficiencies in simulating the dust seasonal cycles, distribution, and optical depth over

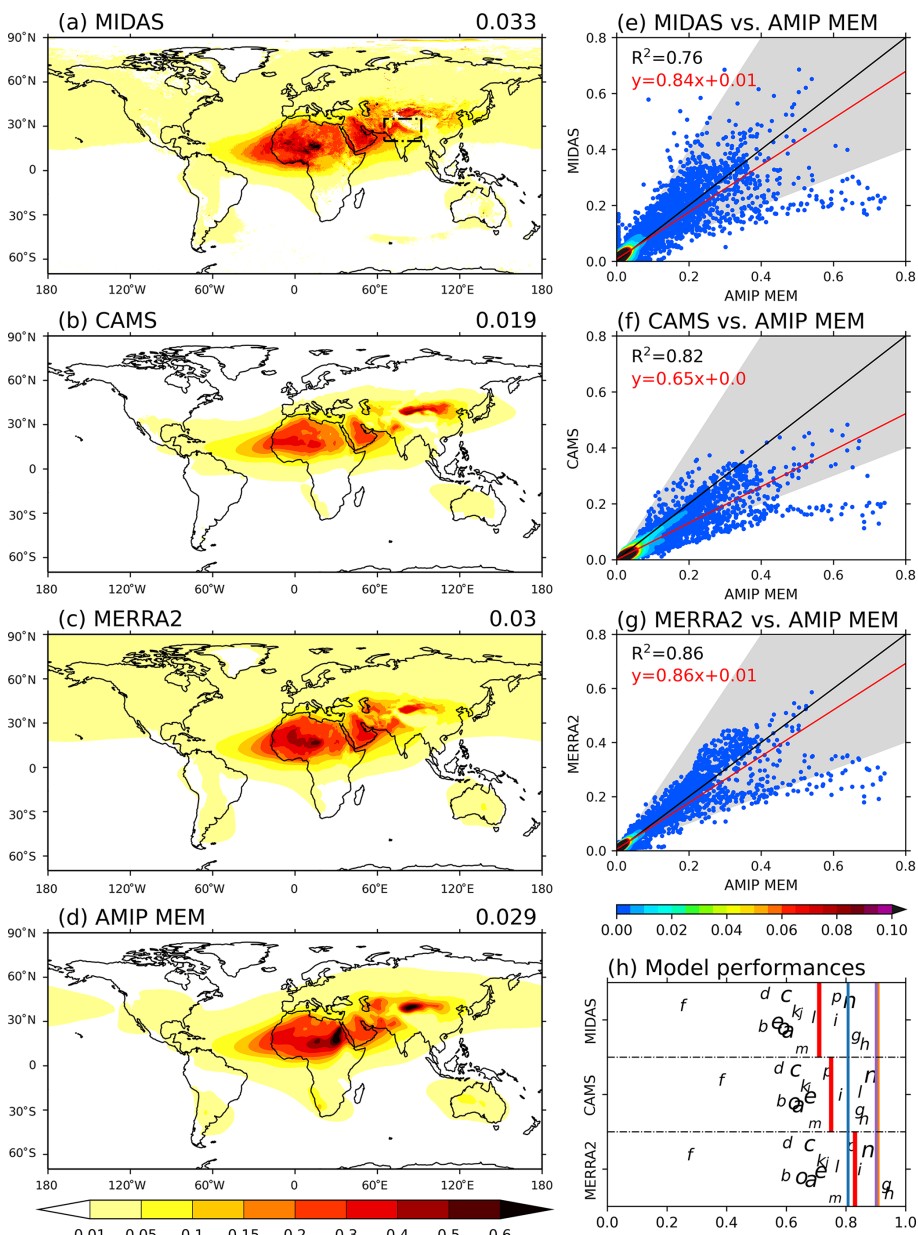

**Figure 13.** Intercomparison of 2005–2014 annual mean DOD from **(a)** MIDAS, **(b)** CAMS, **(c)** MERRA2, and **(d)** AMIP MEM. The black box in **(a)** denotes the region of dust accumulation along the southern slope of the Himalayas. Panels **(e)**–**(g)** show the density ($10^{-4}$) scatterplots of observation/reanalyses (**a–c**; $y$ axes) vs. AMIP MEM (**d**; $x$ axes): the black lines are the 1 : 1 correspondence while the red lines are the linear fitting ($R^2$ and the regression equation given at top left corners). Panel **(h)** is a summary of the performance of each individual model measured by the spatial $R^2$ ($x$ axis) between each individual model and observation/reanalyses. Models are shown by letters (see Table 1). The red vertical bars denote where the AMIP MEM stands. The blue vertical bar shows the spatial $R^2$ between CAMS and MIDAS. Similarly, orange bars are for MERRA2 vs. MIDAS, and purple bars are for CAMS vs. MERRA2.

key source regions. Also, models struggle to agree about the seasonal cycles of dust over a few key source regions such as North China and South and North America. The North China region appears to be particularly challenging for models, which overestimate DOD and frequently represent the seasonal cycle incorrectly in this region.

One limitation of this study is that we were not able to investigate the uncertainties in dust processes associated with the very different assumptions on dust size ranges across models. This is because CMIP6 models only archived particle-size-integrated variables. The difference in dust size ranges might partly explain diversities in models' simulated dust processes such as emission, deposition, load, and life-

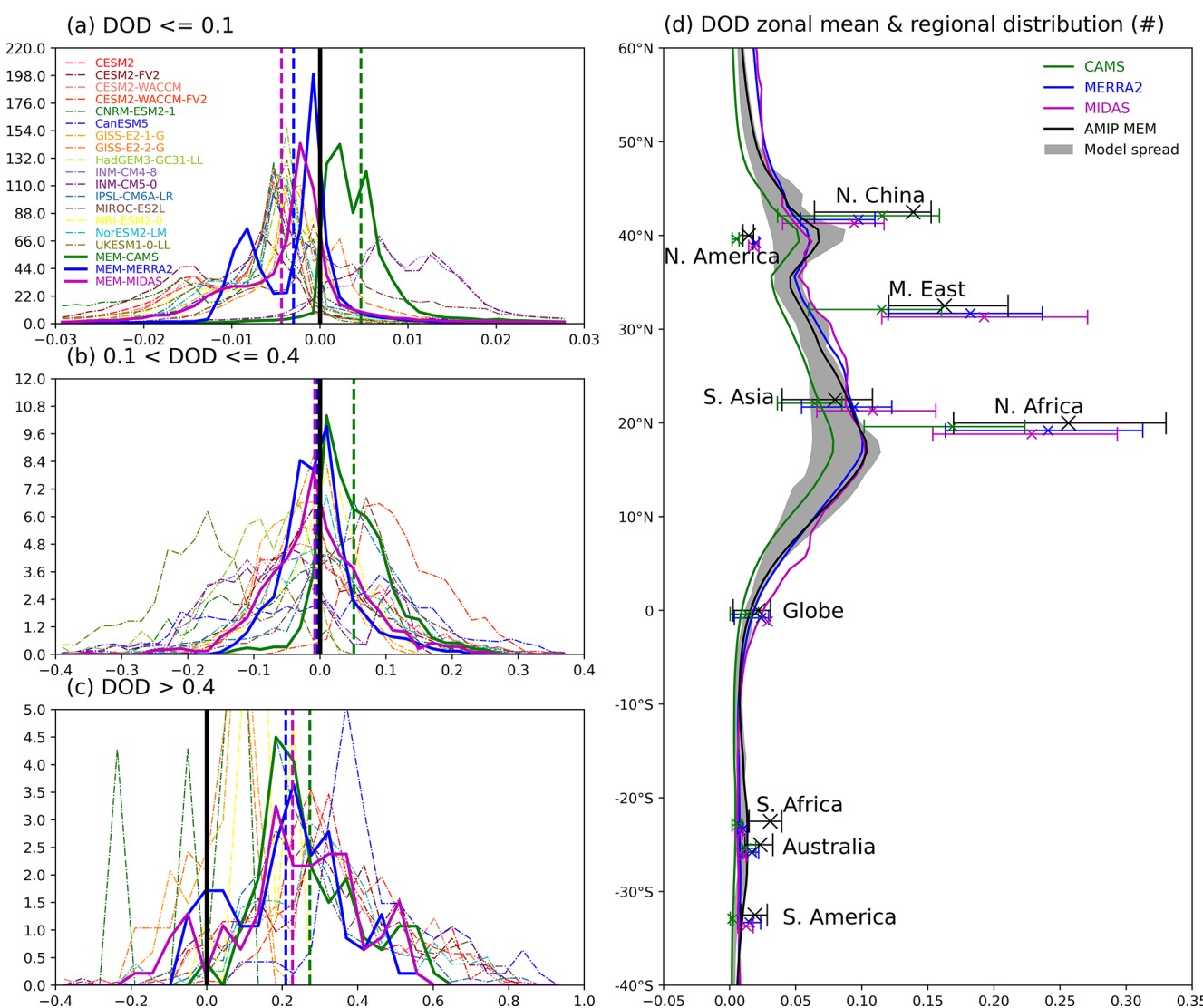

**Figure 14. (a–c)** The frequency (*y* axis in absolute terms) distribution of models to observation/reanalyses DOD differences (*x* axis) in three categories, sampled by a size bin of 0.005. Solid green curves are for AMIP MEM minus CAMS, solid blue curves are for AMIP MEM minus MERRA2, and solid purple curves are for AMIP MEM minus MIDAS. The dashed vertical lines denote the mean of the distributions. The dashed curves represent individual models minus MIDAS. The distribution is calculated using 10 years (2005–2014) of monthly mean data from all grid cells that lie in the corresponding DOD category. The DOD categories refer to AMIP MEM and individual models where applicable. Negative values indicate a model underestimation of DOD. **(d)** Zonal mean DOD profiles (curves; grey shading for 10th–90th percentiles of multi-model spread) and regional mean distributions (error bars for 10th–90th percentile spreads and crosses in the middle for mean values of 2005–2014 annual mean) over the eight dust source regions.

time. We investigated the relationship between maximum particle size represented in each model and global model dust emission, lifetime, and deposition. However, no clear relationship was found (not shown). Therefore, although the maximum size simulated and the transported size distribution clearly contribute to these variables, other model-dependent processes and parameters also contribute. It is therefore challenging to understand the contributions from dust particle maximum size and size distribution without specifically de-

signed experiments and more detailed outputs of dust-related processes such as particle-size-resolved dust fluxes.

The variability across CMIP6 models is generally larger than those in the CMIP5 and AeroCom phase III models. Despite the fact these are different subsets of models with some overlaps, it may indicate that dust processes are becoming more uncertain as models become more sophisticated (Wu et al., 2018). Further, there are inconsistent biases along the life cycle of dust in different models. This indicates the challenges in simulating the links between dust emission, mass

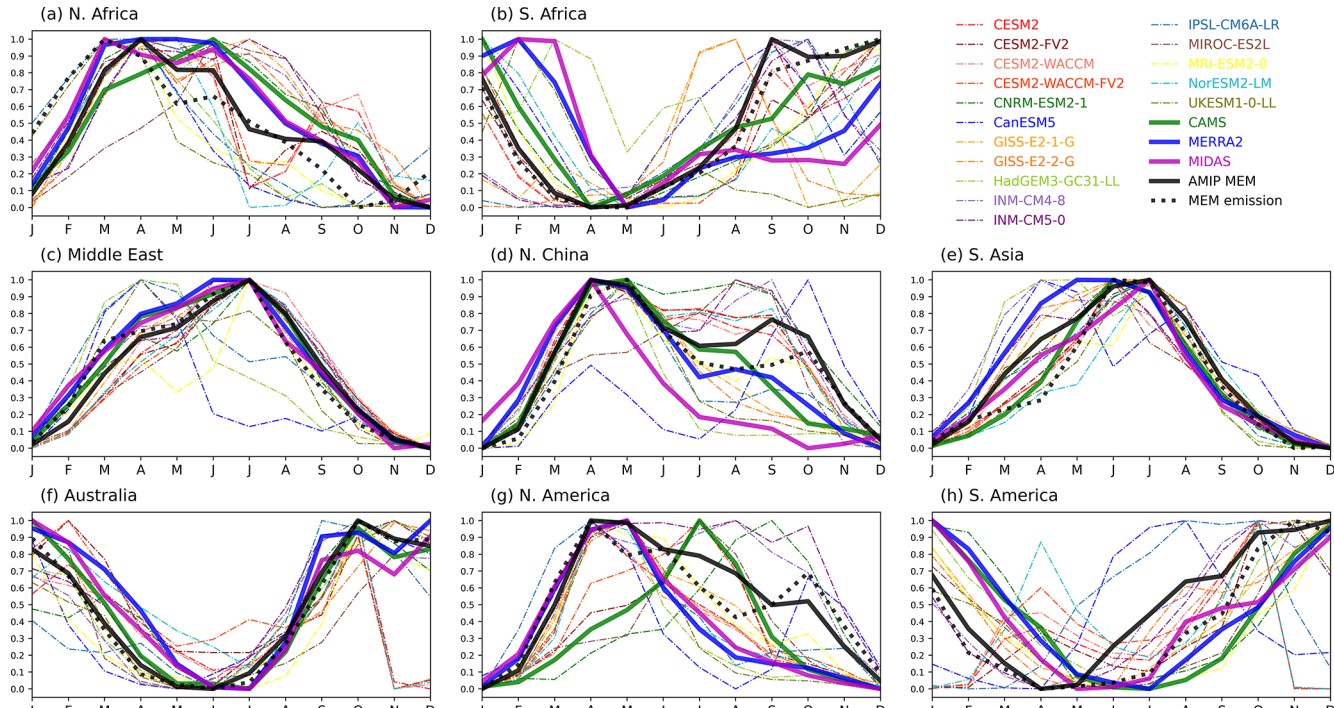

**Figure 15.** Seasonal cycles of DOD over the eight dust source regions. Dashed curves represent individual models, with the AMIP MEM in solid black. Dotted black lines show the AMIP MEM dust emission. Also shown are results from CAMS (solid green), MERRA2 (solid blue), and MIDAS (solid purple). Each curve is normalised against its minimum and maximum. The absolute DOD seasonal cycles are included in Fig. S10.

loading, AOD (relating to size, optical properties), and deposition in CMIP6 models. These challenges come from various sources such as difficulties in accurately simulating dust–land–vegetation–climate interactions. One particular issue is that different models have various assumptions on dust particle size range. This means that diversities in dust sizes, on top of those in model physical processes, add uncertainties to models' simulated dust processes ranging from emission and deposition to mass loading, lifetime, and DOD. We therefore recommend more detailed output relating to the dust cycle in future model intercomparison projects, such as size-resolved dust cycle variables, dust three-dimensional dust burdens, and dust aerosol optical properties. This will enable us to better constrain global dust cycles, as well as the potential identification of observationally constrained links between dust cycles and optical properties. It should be noted that here we examined the AMIP model simulations. It is therefore reasonable to speculate even greater model uncertainties and deficiencies exist in fully coupled CE1 models due to the couplings between dust and many other components of the Earth system.

There are large diversities in CMIP6 models' simulated spatial patterns and magnitudes of DOD. This is particularly true over the dustiest regions such as North Africa, North China, and the Middle East. Almost all models significantly overestimate DOD values over these regions. It is difficult

to investigate the reasons behind this overestimate using the CMIP6 experiments because many dust variables are not available. We suggest that sensitivity experiments may be a better approach to understand this overestimate in future work – one of the outstanding questions to answer in the future centres around the dust particle size distributions over these dust source regions. For example, can model biases in dust particle size distributions explain the overestimation of large DOD values over dust source regions?

It is worthwhile to stress that both the CMIP6 models and the reanalysis datasets fail to capture the spatial patterns of dust over the Middle East, East China, and South Asia. Given that the meteorological drivers (i.e. winds and precipitation) of dust emissions over these regions are largely influenced by large-scale monsoonal circulations, the biases in dust simulations may be ascribed to the poorly represented monsoon systems in CMIP6 models (Wu et al., 2018; Wilcox et al., 2020; Jin et al., 2021). In the meantime, these biases may also cast doubt on model-simulated regional- to global-scale atmospheric circulations and climate states through dust–radiation–climate interactions – for example, the location of the Pacific Intertropical Convergence Zone (ITCZ), which is found to be linearly correlated to dust mass loadings over these regions (Evans et al., 2020).

In summary, the CMIP6 models, and particularly the MEM, generally capture key features of global dust pro-

cesses – for example, the global dust emission regions, global DOD distribution, and dust seasonal cycles over a few key source regions. However, dust aerosols in CMIP6 models still present large uncertainties, and the uncertainty ranges tend to expand compared to previous-generation climate models. The dust processes have inconsistent biases in different models, adding the urgency to better constrain the whole life cycle of dust and the links between different dust processes in climate models. This also provides caveats in interpreting the impacts of dust on other Earth system processes such as the radiation budget, clouds, precipitation, and atmospheric circulations.

**Code and data availability.** This work uses simulations from 16 models participating in the AMIP project as part of the Coupled Model Intercomparison Project (Phase 6; https://www.wcrp-climate.org/wgcm-cmip, World Climate Research Programme, 2020 TS4; Gates et al., 1998); model-specific information can be found through references listed in Table 1. Model outputs are available on the Earth System Grid Federation (ESGF) website (https://esgf-data.dkrz.de/search/cmip6-dkrz/ TS5; Cinquini et al., 2014). Satellite and reanalysis data used in this work are all cited. The analysis was carried out using Bash and Python programming languages.

**Supplement.** The supplement related to this article is available online at: https://doi.org/10.5194/acp-22-1-2022-supplement.

**Author contributions.** AZ, CLR, and LJW conceptualised and designed this study. AZ synthesised and analysed the data, produced the figures, and wrote the paper. CLR and LJW provided insights and comments on the analyses and contributed to the writing of the paper.

**Competing interests.** The contact author has declared that neither they nor their co-authors have any competing interests.

**Disclaimer.** Publisher's note: Copernicus Publications remains neutral with regard to jurisdictional claims in published maps and institutional affiliations.

**Acknowledgements.** We acknowledge the World Climate Research Programme, which, through its Working Group on Coupled Modelling, coordinated and promoted CMIP6. We thank the climate modelling groups for producing and making available their model output, the Earth System Grid Federation (ESGF) for archiving the data and providing access, and the multiple funding agencies who support CMIP6 and ESGF. We thank Ben Johns, Matthew Mizielinkski, Mohit Dalvi, and Jeremy Walton from the UK Met Office for providing model data. We also thank Michela Giusti and Kevin Marsh from the ECMWF for helping to prepare the CAMS dust emission and deposition data. We thank Jasper Kok for providing the compiled dust deposition data. We thank the two anonymous reviewers for their very constructive comments and suggestions that helped to improve this paper.

This work and its contributors Alcide Zhao, Claire L. Ryder, and Laura J. Wilcox were supported by the UK–China Research and Innovation Partnership Fund through the Met Office Climate Science for Service Partnership (CSSP) China as part of the Newton Fund. Claire L. Ryder was supported by a UK Natural Environment Research Council (NERC) independent research fellowship grant (grant no. NE/M018288/1).

**Financial support.** This research has been supported by the UK–China Research and Innovation Partnership Fund through the Met Office Climate Science for Service Partnership (CSSP) China as part of the Newton Fund and the Natural Environment Research Council (NERC) (grant no. NE/M018288/1). TS6

**Review statement.** This paper was edited by Ashu Dastoor and reviewed by two anonymous referees.

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

## Remarks from the language copy-editor

CE1  I'm not sure if you requested a change here or if it was just an accidental mark. If you did request the hyphenation of "fully coupled", note that this is correct without a hyphen as we do not hyphenate –ly adverbs (Chicago pp. 227–228, 373) as ambiguity is virtually impossible (Chicago, p. 374).

## Remarks from the typesetter

TS1  Please note that the initials "M." and "C." cannot be removed as they differentiate the two citations of Wu et al., 2020 (please also see remark TS47 of the previous proofreading).

TS2  Please confirm change to Gelaro according to the reference list entry you provided in the first proofreading.

TS3  According to our standards, all changes in values must first be approved by the editor, as data have already been reviewed, discussed and approved. Please provide a detailed explanation for those changes that can be forwarded to the editor. Please note that this entire process will be available online after publication. Upon approval, we will make the appropriate changes. Thank you for your understanding.  74% was a wrong citation to the one of the number in Table 2; it should be 63%

TS4  Please provide the corresponding reference list entry for the citation "World Climate Research Programme, 2020".

TS5  Please provide a reference list entry including creators, title, repository/publisher, and date of last access.

TS6  Thank you for providing this information. Please note that we allow the funding information to be included in both the acknowledgements and the financial support section if you would like to leave the acknowledgements section as it is, or this information can now be removed from the acknowledgements. Please let us know how you would like to proceed. Thank you.

TS7  Please confirm article number as provided by the articles website.

TS8  Please provide all editor names in the correct order (last name, initials).

TS9  Please provide journal.

TS10  Please provide journal (according to the website, it is J. Geophys. Res.-Atmos. and not Geoscientific Model Development).

TS11  Please provide journal.

TS12  Please provide journal and page range or article number.