# Peer review of "How well do the CMIP6 models simulate dust aerosols?"

_Atmospheric Chemistry and Physics, 2021_

## Referee Comment (RC1)

Review of "How well do the CMIP6 models simulate dust aerosols?" by Alcide Zhao, Claire L. Ryder, and Laura J. Wilcox

This paper examines the performance of 16 CMIP6 models in simulating dust emissions, deposition, burden, lifetime, and dust optical depth (DOD) for the present day climatology (2005-2014). The AMIP-type of model results are compared with reanalyses (CAMS and MERRA-2) and satellite retrievals. It is found that models in general capture the spatial patterns of dust emissions, loading, and deposition but have discrepancies in representing seasonal cycles of dust over North China, North America, and South America and show a large diversity in simulating DOD. It's a timely paper providing an informative evaluation of the life cycle of dust in CMIP6 models. While I appreciate the authors' great work and detailed analysis, I found the uncertainties of the data and comparisons are not well addressed in the paper. My comments are listed as follows.

**Major comments:**

1. While aerosol reanalysis products, e.g., CAMS and MERRA-2, provide high spatial and temporal resolutions of data to study dust, it should be noted that variables such as dust emission, deposition, dust loading, and DOD, are not directly constrained by observations—only AOD is directly assimilated with satellite retrievals and ground observations. The uncertainties associated with reanalysis variables and how these may affect the comparison with CMIP6 model output need to be discussed.

I'm also not fully convinced by the argument that "we did not evaluate the models against ground-based measurements but instead perform a large-scale analysis focusing on the more spatio-temporally available fields from reanalyses that are in good agreement with ground-based observations where they exist (Wu et al., 2020a, b)." (lines 347-349). While MERRA-2 may generally have a good agreement with ground-based deposition data as noted by previous work, it does not necessarily imply that ground-based measurements can be substituted by reanalysis products. Not to mention ground-based deposition data also have uncertainties.

In short, a clarification and detailed discussion of the uncertainties of the data used here as references for CMIP6 model evaluation would be useful. For satellite products, e.g., MIDAS DOD and FMI AOD, providing error ranges, which can be found in the referred papers in the text, would be helpful, too.

2. How the timing of dust emissions affects the seasonal peak of DOD is briefly discussed for North Africa in Fig. 13 (lines 337-339). I wonder if it's possible to add more discussion about generally how the representation of dust emissions and deposition affect model performance in simulating DOD.

Minor points:

1. Line 138, how is bare soil fraction defined? Do you use the output of bare soil fraction from each model to perform the regression?

2. Section 2.4, a similar multiple linear regression was used by Pu and Ginoux (2018) to study drivers of global DOD.

3. Line 167, it would be better to add the definitions of regions (boxes) to Fig. 1 instead of keeping it in the supplement (Fig. S1).

4. Line 175, please consider adding a discussion about the uncertainties associated with dust emissions from CAMS and MERRA-2.

5. Line 194, have you compared dust emissions for CMIP6 models with and without dynamic vegetation schemes to see if the latter generally have larger uncertainties and emissions?

6. Line 197, "28-69", is this a range from the  $10^{th}$  to  $90^{th}$  percentile or the minimum to maximum?

7. Line 245, cannot find any clear impact of soil bareness (brown shading) from CAMS in Fig. 4I.

8. Lines 274-275, very interesting. Any idea about why? e.g., do you have a figure similar to Fig. 7 to clarify over what regions the wet deposition are high in CESM2?

9. Lines 281-282, this adds to the need to discuss the uncertainties of evaluating CMIP6 model output against CAMS and MERRA-2.

10. Line 319, "Fig. 11a", referred to a wrong figure?

11. Line 329, "black crosses", refers to model mean?

12. Lines 363, "dust accumulations along the southern edge of the Himalayas" can you please highlight that part in the figure? And what about uncertainties associated with satellite revivals of dust from snow surface?

13. Fig. 2 caption, "The global annual total dust emission budgets (e; Pg yr-1)", I think you refer to Fig. 2(d), right?

14. Fig. 6, consider adding boxes of the meridionally-averaged regions, Asia and Africa, to plot 6(a).

15. Fig. 8, maybe use a boxplot or add error bars (e.g., the 10th to 90th percentiles) for AMIP MEM instead of using a plus sign?

16. Fig. 9, why does the font size of the letters in (h) differ? What is spatial  $R^2$ ? Is it pattern correlation?

17. Fig. 12(a)-(c), what is the x-axis? Is the y-axis frequency (%)?

18. Fig. 13, maybe add the lines of AMIP MEM dust emissions for a better comparison of the timing of emission and DOD maxima?

---

## Author Comment (AC1)

The authors are grateful to the two referees for their interest and comments on the paper. These comments are very valuable and have helped improve the manuscript. Here we outline how we have addressed these comments in the revised manuscript. In this response, blue parts are our replies/explanations to each comment, and red texts are changes made in the manuscript.
The revised version of the manuscript with changes tracked is attached separately.

**Replies to Referee #1:**

Major comments:
1. While aerosol reanalysis products, e.g., CAMS and MERRA-2, provide high spatial and temporal resolutions of data to study dust, it should be noted that variables such as dust emission, deposition, dust loading, and DOD, are not directly constrained by observations—only AOD is directly assimilated with satellite retrievals and ground observations. The uncertainties associated with reanalysis variables and how these may affect the comparison with CMIP6 model output need to be discussed.
I'm also not fully convinced by the argument that "we did not evaluate the models against ground-based measurements but instead perform a large-scale analysis focusing on the more spatio-temporally available fields from reanalyses that are in good agreement with ground-based observations where they exist (Wu et al., 2020a, b)." (lines 347-349). While MERRA-2 may generally have a good agreement with ground-based deposition data as noted by previous work, it does not necessarily imply that ground-based measurements can be substituted by reanalysis products. Not to mention ground-based deposition data also have uncertainties.
In short, a clarification and detailed discussion of the uncertainties of the data used here as references for CMIP6 model evaluation would be useful. For satellite products, e.g., MIDAS DOD and FMI AOD, providing error ranges, which can be found in the referred papers in the text, would be helpful, too.

We thank the reviewer for their very insightful comments and suggestions.
In response to this comment in combination with comments from Reviewer #2, we have taken the following actions:
  1) In the abstract, we have added a sentence "We note that both the reanalyses and observations used here have their limitations, and particularly that dust emission and deposition in reanalyses are poorly constrained."
  2) In Sec. 2.2, following the sentence "MIDAS was calculated using quality filtered MODIS-Aqua AOD retrievals along with DOD-to-AOD ratios provided by the Modern-Era Retrospective analysis for Research and Applications version 2 reanalysis (MERRA2)", we added "This means that the MIDAS DOD estimates are also model-dependent and uncertain."
  3) In Sec. 2.3, we added a new paragraph following the descriptions of the two reanalysis datasets: "It is important to note that only AOD from observations is assimilated in CAMS and MERRA2. The DOD and dust mass loading are then adjusted based on the contribution of DOD to AOD, which will vary in space and time. Therefore, the accurate representation of DOD and dust mass loading in the reanalyses rely on the simulation of correct proportions of dust relative to other aerosol species. While this aerosol speciation may be well represented in locations or time periods dominated by dust (e.g. over the

remote Sahara), it is likely to be less well represented in regions where different aerosol species coexist (e.g. over northern India, with mixed dust, smoke and anthropogenic aerosol). Additionally, the reanalyses adjust DOD and dust mass loading via data assimilation, but this will not be fed through to changes in dust emission, which remain an unconstrained model variable in the reanalyses. This means that despite the assimilation of satellite AOD retrievals, dust processes in CAMS and MERRA2 remain model-dependent and entail some level of uncertainty (Xian et al., 2020). Therefore, the comparisons between models and the reanalyses presented here should be interpreted with some caution."

4) In sec. 4, we added another bullet point to stress this point:
"Dust processes in the CAMS and MERRA2 are very uncertain, as demonstrated by the twofold difference in global dust load (23 Tg in MERRA2 vs 12 Tg for CAMS)."

In response to the reviewer's suggestion on comparing models with ground observations, we attempted to compare models' simulated surface dust concentration and total dust depositions to those compiled by Albani et al. (2014). Unfortunately, only four models (CNRM-ESM2-1, GISS-E2-1-G, GISS-E2-2-G, MRI-ESM2-0) have output for surface dust concentration (CMIP6 variable: sconcdust). We therefore decided to focus on evaluating total dust deposition in this work. The results are shown in Fig. 9 (copied below) and Supplementary Fig. S8 (individual models).

Fig.9 shows that CAMS and MERRA2 give fair representations of dust deposition compared to the observations; this also justifies the use of CAMS and MERRA2 in this paper. It also shows that the MEM is as good as the reanalyses in simulating dust deposition.

We add the following in Section 3.2 to comment on this: "The intercomparisons between reanalyses, models, and ground observations of total dust deposition fluxes (Figure 9) show that CAMS and MERRA2 give fair representations of dust deposition compared to the observations (i.e., with a log-space root mean square error (RMSE) of ~2.0). Meanwhile, the MEM and most individual models (Figure S8) are as good as reanalyses. We note however the observational dust deposition fluxes only include $PM_{10}$ particles, while models and reanalyses datasets have larger dust particles. Therefore, whilst biases are expected, we are not able to quantify them due to the lack of size-resolved dust deposition fluxes from models in the CMIP6 archive."

We have also added in the text the uncertainties related to satellite observed global mean AOD and DOD. For example: "CMIP6 model-simulated global mean DOD varies by a factor of 7 from 0.011 to 0.073, with the MEM estimate of 0.029. This is consistent with observationally constrained estimate of 0.030±0.005 for $PM_{20}$ dust (Ridley et al., 2016), 0.033 (0.031-0.040) in MIDAS, 0.031 (0.028-0.036) in MERRA2, but is ~1.5 times that of CAMS (~0.019)."

[Figure]

Figure 9: Scatterplots of annual mean total dust deposition flux at ground stations between (a) MEERA2 and CAMS, (b) AMIP MEM and Observations, (c) CAMS and observations, and (d) MERRA2 and observations. The stations are marked with different styles and colours for different locations (cf. Figure 1). The correlation coefficients and root mean square errors (RMSE) are calculated in log space. The 1:1 (solid) and 1:10/10:1 (dotted) lines are plotted for reference. The scatter plots between each individual model and the observations can be found in Supplementary Figure 8.

2. How the timing of dust emissions affects the seasonal peak of DOD is briefly discussed for North Africa in Fig. 13 (lines 337-339). I wonder if it's possible to add more discussion about generally how the representation of dust emissions and deposition affect model performance in simulating DOD.

We thank the reviewer for their suggestion. We have reproduced Fig. 15 (copied below), with the MEM dust emission seasonal cycles added as dotted black lines, to investigate this.

We found that DOD seasonal cycles follow dust emission in all other regions. We have rewritten the sentence into "Finally, it is interesting to note that the seasonal cycle of DOD over North Africa (Figure 15a) peaks slightly later than dust emission. This may indicate the importance of dust transport in influencing dust optical depth and its seasonal cycles. In comparison, the seasonal cycles of DOD are synchronised with dust emissions in MEM over all other regions. Therefore, model biases in dust emissions are likely to be reflected in DOD."

Minor points:
1. Line 138, how is bare soil fraction defined? Do you use the output of bare soil fraction from each model to perform the regression?
The reviewer is correct that we are using the model output (baresoilFrac: see here: http://clipc-services.ceda.ac.uk/dreq/u/baa84fd4-e5dd-11e5-8482-ac72891c3257.html).
baresoilFrac is defined as "'bare soil percentage area coverage"

2. Section 2.4, a similar multiple linear regression was used by Pu and Ginoux (2018) to study drivers of global DOD.

We thank the reviewer for noting this, and have cited this paper in the revised version.

[Figure]

**Figure 15**: Seasonal cycles of DOD over the eight dust source regions. Dashed curves represent individual models, with the AMIP MEM in solid black. Dotted black lines show the AMIP MEM dust emission. Also shown are results from CAMS (solid green), MERRA2 (solid blue) and MIDAS (solid purple). Each curve is normalized against its minimum and maximum. The absolute DOD seasonal cycles are included in Supplementary Figure S10.

3. Line 167, it would be better to add the definitions of regions (boxes) to Fig. 1 instead of keeping it in the supplement (Fig. S1).
We have added a new Figure 1 (copied below) to define regions, together with the locations of ground dust deposition measurement data we used to evaluate the models.

[Figure]

**Figure 1**: The CMIP6 AMIP MEM-simulated 2005-2014 annual mean dust emission (g m$^{-2}$ yr$^{-1}$) climatology overlaid by boxes used to define major dust emission source regions. The coloured symbols denote groupings of observations by different regions following Kok et. al., (2021).

4. Line 175, please consider adding a discussion about the uncertainties associated with dust emissions from CAMS and MERRA-2.

We thank the reviewer for their very insightful comments and suggestions.
In response to this comment in combination with comments from Reviewer #2, we have taken the following actions:

1) In the abstract, we have added a sentence "We note that both the reanalyses and observations used here have their limitations, and particularly that dust emission and deposition in reanalyses are poorly constrained."

2) In Sec. 2.2, following the sentence "MIDAS was calculated using quality filtered MODIS-Aqua AOD retrievals along with DOD-to-AOD ratios provided by the Modern-Era Retrospective analysis for Research and Applications version 2 reanalysis (MERRA2).", we added "This means that the MIDAS DOD estimates are also model-dependent and uncertain."

3) In Sec. 2.3, we added a new paragraph following the descriptions of the two reanalysis datasets: "It is important to note that only AOD from observations are assimilated in CAMS and MERRA2. The DOD and dust mass loading are then adjusted based on the contribution of DOD to AOD, which will vary in space and time. Therefore, the accurate representation of DOD and dust mass loading in the reanalyses rely on the simulation of correct proportions of dust relative to other aerosol species. While this aerosol speciation may be well represented in locations or time periods dominated by dust (e.g. over the remote Sahara), it is likely to be less well represented in regions where different aerosol species coexist (e.g. over northern India, with mixed dust, smoke and anthropogenic aerosol). Additionally, the reanalyses adjust DOD and dust mass loading via data assimilation, but this will not be fed through to changes in dust emission, which remain an unconstrained model variable in the reanalyses. This means that despite the assimilation of satellite AOD retrievals, dust processes in CAMS and MERRA2 remain model-dependent and entail some level of uncertainty (Xian et al., 2020). Therefore, the comparisons between models and the reanalyses presented here should be interpreted with some caution."

4) In sec. 4, we added another bullet point to stress this point:
"Dust processes in the CAMS and MERRA2 are very uncertain, as demonstrated by the twofold difference in global dust load (23 Tg in MERRA2 vs 12 Tg for CAMS)."

5. Line 194, have you compared dust emissions for CMIP6 models with and without dynamic vegetation schemes to see if the latter generally have larger uncertainties and emissions?

We thank the reviewer for the great point which was one of the initial goals of this study. However, unfortunately, we were not able to compare models with and without dynamic vegetations due to the low numbers of model with dynamic vegetation performing the required simulations.

6. Line 197, "28-69", is this a range from the 10th to 90th percentile or the minimum to maximum?

We thank the reviewer for their careful read and apologise for the confusion. These numbers refer to the min-max range. We have now clarified this in the revised version. The text now reads: "North Africa contributes the most (57 (minimum-maximum: 28-69) %) to global dust emissions in CMIP6 models (Figure 3d),

generally agreeing with CAMS (46%) and MERRA2 (60%) …"

7. Line 245, cannot find any clear impact of soil bareness (brown shading) from CAMS in Fig. 4I.
The reviewer is correct that surface wind dominates in CAMS and shading (contribution) from soil bareness cannot be seen. However, surface bareness does play a role next to surface wind in CAMS. We now demonstrate this with a new supplementary Fig. S2 (copied below) showing drivers of emission specifically for CAMS.

[Figure]

**Figure S2** Normalised relative importance (left axis) of the three major dust emission drivers throughout the year over the eight major source regions in CAMS. Purple for precipitation, blue for surface wind speed, and brown for bare soil fraction. The black curves CAMS seasonal cycles of dust emissions (right axis; mg m$^{-2}$ day$^{-1}$).

8. Lines 274-275, very interesting. Any idea about why? e.g., do you have a figure similar to Fig. 7 to clarify over what regions the wet deposition are high in CESM2?
As summarised in Table 2, and the new supplementary Fig. S7 (copied below), the wet depositions are too high over oceans in the CESM2 family models.
The text now reads: "yet the CESM2 models show that most (~74%) of the total dust removal is via wet processes (also see Figure S7)."

[Figure]

Figure S7: Comparisons of 2005-2014 mean of annual total (dry + wet) dust deposition (left; g m-2 yr-1) and the ratio of wet-to-total depositions (right; %) in the CESM2 family models. The numbers on the top right of each panel denote the global total dust deposition flux (Tg yr-1) and the fraction of global wet-to-total dust depositions (%).

9. Lines 281-282, this adds to the need to discuss the uncertainties of evaluating CMIP6 model output against CAMS and MERRA-2.
We thank the reviewer for their very insightful comments and suggestions. In response to this comment in combination with comments from Reviewer #2, we have taken the following actions:
1) In the abstract, we have added a sentence "We note that both the reanalyses and observations used here have their limitations, and particularly that dust emission and deposition in reanalyses are poorly constrained."
2) In Sec. 2.2, following the sentence "MIDAS was calculated using quality filtered MODIS-Aqua AOD retrievals along with DOD-to-AOD ratios provided by the Modern-Era Retrospective analysis for Research and Applications version 2 reanalysis (MERRA2).", we added "This means that the MIDAS DOD estimates are also model-dependent and uncertain."
3) In Sec. 2.3, we added a new paragraph following the descriptions of the two reanalysis datasets: "It is important to note that only AOD from observations are assimilated in CAMS and MERRA2. The DOD and dust mass loading are then adjusted based on the contribution of DOD to AOD, which will vary in space and time. Therefore, the accurate representation of DOD and dust mass loading in the reanalyses rely on the simulation of correct proportions of

dust relative to other aerosol species. While this aerosol speciation may be well represented in locations or time periods dominated by dust (e.g. over the remote Sahara), it is likely to be less well represented in regions where different aerosol species coexist (e.g. over northern India, with mixed dust, smoke and anthropogenic aerosol). Additionally, the reanalyses adjust DOD and dust mass loading via data assimilation, but this will not be fed through to changes in dust emission, which remain an unconstrained model variable in the reanalyses. This means that despite the assimilation of satellite AOD retrievals, dust processes in CAMS and MERRA2 remain model-dependent and entail some level of uncertainty (Xian et al., 2020). Therefore, the comparisons between models and the reanalyses presented here should be interpreted with some caution."

4) In sec. 4, we added another bullet point to stress this point:
"Dust processes in the CAMS and MERRA2 are very uncertain, as demonstrated by the twofold difference in global dust load (23 Tg in MERRA2 vs 12 Tg for CAMS)."

10. Line 319, "Fig. 11a", referred to a wrong figure?
No – this is the right figure: we referred to the DOD pattern over N. India (Fig. 13a in the revised version).

11. Line 329, "black crosses", refers to model mean?
We thank the reviewer for pointing the confusion out. Yes, it is the multi-model mean which is now clarified in the revised version.

12. Lines 363, "dust accumulations along the southern edge of the Himalayas" can you please highlight that part in the figure? And what about uncertainties associated with satellite revivals of dust from snow surface?
We have now added a box in the revised version Fig. 13a.
The dust referred to accumulates in the atmosphere just to the south of the Himalayas, therefore contributions to uncertainty in DOD retrievals are extremely minor since this region is generally not snow covered. Over the Himalayan mountain range, dust optical depths are extremely low (as expected), so that the mountain range acts as a natural barrier trapping dust (and other) aerosols over the Indo-Gangetic Plains. This can be seen in the MIDAS data in figure 11a, and is also found in other datasets (e.g. Pu & Ginoux 2018 for MODIS DOD and CALIOP DOD). Sayer et al. (2019) show that MODIS underestimates AOD compared to AERONET in this region, and similarly, Gikkas et al (2021) show that the region of the Thar desert MIDAS DOD was 0.125 compared to 0.169 from the LIVAS dataset. The main uncertainty in MIDAS DOD here is likely to arise from incorrect classification of aerosol type.
We have re-worded this bullet point more clearly, however:
"The CMIP6 models consistently fail to capture certain key features of regional dust distributions. For example, atmospheric dust accumulation to the south of the Himalayas over the Indo-Gangetic Plains, and regional DOD variability over East and Central Asia and the Middle East."

13. Fig. 2 caption, "The global annual total dust emission budgets (e; Pg yr-1)", I think you refer to Fig. 2(d), right?

We thank the reviewer for their careful read. Yes, it should be panel d, which is corrected in the revised version.

14. Fig. 6, consider adding boxes of the meridionally-averaged regions, Asia and Africa, to plot 6(a).

We thank the reviewer for the suggestion. Boxes are now added in the revised Fig. 7c (copied below).

[Figure]

**Figure 7**: Intercomparison of 2005-2014 annual mean dust mass loading (mg m$^{-2}$) between (a) AMIP MEM, (b) CAMS and (c) MERRA2. The numbers on the top right of each panel denote the global total dust burden (Tg). Maps for individual models can be found in Supplementary Figure 3. (d) Global total dust burden from each individual model as well as those of (a-c): boxes denote the 10$^{th}$-90$^{th}$ percentiles of the annual variability; red pluses denote outliers that are outside 1.5 times of annual standard deviation. The vertical pink shading represents the 10$^{th}$-90$^{th}$ percentiles of the multimodal spread. Also shown are the meridionally-averaged DOD over (e) the Africa-Atlantic region (0-35N, 60W-0W; box in (c)) in June-July-August and (f) the Asia-Pacific region (10-40N, 100E-150E; box in (c))) in April-May-June.

15. Fig. 8, maybe use a boxplot or add error bars (e.g., the 10th to 90th percentiles) for AMIP MEM instead of using a plus sign?
Good idea – the 10$^{th}$-90$^{th}$ percentile error-bars are now added in the revised Figure 10 (copied below).

16. Fig. 9, why does the font size of the letters in (h) differ? What is spatial R2 ? Is it pattern correlation?
We apologise for the visual confusion, but the font sizes are the same in the new Fig. 11h and Fig.13h.
R$^2$ refers to the spatial correlation, which is now clarified in the figure captions.

[Figure]

**Figure 10**: Scatterplots of (a) global annual mean total dust burdens (Tg) vs. annual total dust deposition (Pg yr$^{-1}$), and global dust lifetime (days) vs. (b) the ratio of global dry-to-total deposition (%), (c) total dry depositions (Pg yr$^{-1}$), and (d) total wet depositions (Pg yr$^{-1}$). Model colour codes are the same as in other figures, along with CAMS (green cross) and MERRA2 (blue cross). The AMIP multi-model mean and spread (10[th]-90[th] percentiles) are shown by black pluses. The dotted slope lines in (a) denote dust lifetime intervals (days). The solid slope lines in (b) and (c) are the linear fitting between X and Y axis using all data points. All results shown are 2005-2014 annual mean.

17. Fig. 12(a)-(c), what is the x-axis? Is the y-axis frequency (%)?
The X-axis, as explained in Fig. 12 caption, is the differences between observation/reanalyses and models. Y-axis is the frequency in absolute terms.
We have further clarified this in Figure 14 caption: "The frequency (Y-axis in absolute terms) distribution of models to observation/reanalyses DOD differences (X-axis) in three categories, …"

18. Fig. 13, maybe add the lines of AMIP MEM dust emissions for a better comparison of the timing of emission and DOD maxima?
We thank the reviewer for the great suggestion, and have added the AMIP MEM dust emissions (black dotted lines) in the revised Fig. 15 (copied below).

[Figure]

**Figure 15**: Seasonal cycles of DOD over the eight dust source regions. Dashed curves represent individual models, with the AMIP MEM in solid black. Dotted black lines show the AMIP MEM dust emission. Also shown are results from CAMS (solid green), MERRA2 (solid blue) and MIDAS (solid purple). Each curve is normalized against its minimum and maximum. The absolute DOD seasonal cycles are included in Supplementary Figure S10.

**Replies to Referee #2:**

Major Comments:

1. The MERRA2 and CAMS reanalysis products have substantial biases, as evidenced by the twofold contrast of global load (23 Tg for MERRA2 v. 12 Tg for CAMS: line 259). Both reanalysis products rely upon models that assimilate total AOD. The problem is that the contribution of dust to total AOD (the dust optical depth or DOD) is strongly model-dependent. The reanalysis models, like the ESMs themselves, make a number of assumptions. The article notes that both reanalyses compute emission using a scheme taken from Ginoux et al. (2001). There are many admirable features of the Ginoux study, but the calculation of emission has a particle size dependence that is now recognized to give unphysical emphasis to smaller particles (as discussed by Legrand et al. GMD 2019 https://doi.org/10.5194/gmd-12-131-2019). Moreover, both models weight emission using the Ginoux topographic source map. Other regional weightings (i.e. erodibility maps) are used in some ESMs. These maps are equally plausible but emphasize different regional sources. (See Fig.1 of Cakmur et al. JGR 2006.) In summary, the MERRA2 and CAMS reanalyses are heavily dependent upon modeling assumptions, just like the ESMs, which undermines the use of the reanalyses as an observational standard. There is a nice comparison of MERRA2 and CAMS by Xian and Klotzbach et al. (ACP 2020 https://doi.org/10.5194/acp-20-15357-2020), who instead recommend a multi-reanalysis composite, while emphasizing the resulting uncertainty. The authors of the present article also use the ModIs Dust AeroSol (MIDAS) product for ESM evaluation. The problem is that they construct DOD given the retrieved AOD combined with the MERRA2 ratio of these two variables. In other words, the "observed" DOD in fact is dependent upon the contribution of dust compared to the total aerosol extinction as calculated by a single model. In the end, the effect of differing reanalysis model assumptions means that their output is a highly uncertain standard with limited influence of actual observations. (Again, as evidence, note the difference in global load between the reanalyses.)

In fairness to the authors, one challenge of evaluating a dust model is that instrument retrievals, which are spatially detailed and available for as long as two decades, do not differentiate between different aerosol species. Their use introduces uncertainty into any evaluation of a model dust cycle. However, the reanalyses are highly uncertain for the same reason. The authors need to address this uncertainty rather than just talk about model diversity and biases. (Neither reanalysis product is named in the abstract nor are any of their disagreements noted.) The uncertainty and lack of consensus among the reanalyses has to be an explicit part of the study and given emphasis in the abstract and conclusions.

We thank the reviewer for their very insightful comments and suggestions.
In response to this comment in combination with comments from Reviewer #2, we have taken the following actions:
1) In the abstract, we have added a sentence "We note that both the reanalyses and observations used here have their limitations, and particularly that dust emission and deposition in reanalyses are poorly constrained."
2) In Sec. 2.2, following the sentence "MIDAS was calculated using quality filtered MODIS-Aqua AOD retrievals along with DOD-to-AOD ratios provided by the Modern-Era Retrospective analysis for Research and Applications version 2 reanalysis (MERRA2).", we added "This means that the MIDAS DOD estimates are also model-dependent and uncertain."

3) In Sec. 2.3, we added a new paragraph following the descriptions of the two reanalysis datasets: "It is important to note that only AOD from observations are assimilated in CAMS and MERRA2. The DOD and dust mass loading are then adjusted based on the contribution of DOD to AOD, which will vary in space and time. Therefore, the accurate representation of DOD and dust mass loading in the reanalyses rely on the simulation of correct proportions of dust relative to other aerosol species. While this aerosol speciation may be well represented in locations or time periods dominated by dust (e.g. over the remote Sahara), it is likely to be less well represented in regions where different aerosol species coexist (e.g. over northern India, with mixed dust, smoke and anthropogenic aerosol). Additionally, the reanalyses adjust DOD and dust mass loading via data assimilation, but this will not be fed through to changes in dust emission, which remain an unconstrained model variable in the reanalyses. This means that despite assimilation of satellite AOD retrievals, dust processes in CAMS and MERRA2 remain model-dependent and entail some level of uncertainty (Xian et al., 2020). Therefore, the comparisons between models and the reanalyses presented here should be interpreted with some caution."

4) In sec. 4, we added another bullet point to stress this point:
"Dust processes in the CAMS and MERRA2 are very uncertain, as demonstrated by the twofold difference in global dust load (23 Tg in MERRA2 vs 12 Tg for CAMS)."

In response to the reviewer's suggestion on comparing models with ground observations, we attempted to compare models' simulated surface dust concentration and total dust depositions to those compiled by Albani et al. (2014). Unfortunately, only four models (CNRM-ESM2-1, GISS-E2-1-G, GISS-E2-2-G, MRI-ESM2-0) have output for surface dust concentration (CMIP6 variable: sconcdust). We therefore decided to focus on evaluating total dust deposition in this work. The results are shown in Fig. 9 (copied below) and Supplementary Fig. S8 (individual models).

Fig.9 shows that CAMS and MERRA2 give fair representations of dust deposition compared to the observations; this also justifies the use of CAMS and MERRA2 in this paper. It also shows that the MEM is as good as the reanalyses in simulating dust deposition.

We add the following in Section 3.2 to comment on this: "The intercomparisons between reanalyses, models, and ground observations of total dust deposition fluxes (Figure 9) show that CAMS and MERRA2 give fair representations of dust deposition compared to the observations (i.e., with a log-space root mean square error (RMSE) of ~2.0). Meanwhile, the MEM and most individual models (Figure S8) are as good as reanalyses. We note however the observational dust deposition fluxes only include $PM_{10}$ particles, while models and reanalyses datasets have larger dust particles. Therefore, whilst biases are expected, we are not able to quantify them due to the lack of size-resolved dust deposition fluxes from models in the CMIP6 archive."

[Figure]

Figure 9: Scatterplots of annual mean total dust deposition flux at ground stations between (a) MEERA2 and CAMS, (b) AMIP MEM and Observations, (c) CAMS and observations, and (d) MERRA2 and observations. The stations are marked with different styles and colours for different locations (cf. Figure 1). The correlation coefficients and root mean square errors (RMSE) are calculated in log space. The 1:1 (solid) and 1:10/10:1 (dotted) lines are plotted for reference. The scatter plots between each individual model and the observations can be found in Supplementary Figure 8.

2. Another problem with the article is its limited consideration of particle size range. While the authors note that "dust particle size range represented differ significantly between models" (line 102), their discussion of diversity of model emission does not account for this varying range. In the Abstract (line 10), they write "For example, global dust emissions, primarily driven by model-simulated surface winds, vary by a factor of 5 across models, while the MEM estimate is double the amount in reanalyses. " Not all of this diversity is a result of uncertain representation of the physical processes controlling emission. Some of it is simply based on a somewhat arbitrary decision by each modeling group about the maximum particle size to represent. The authors cite a multi-model ensemble mean (MEM) emission of 3.5 Pg/yr (line 189), but this average results from the combination of emission from models with varying size ranges and does not solely reflect our imprecise knowledge of emission physics. On line 191, the authors refer to the "observationally-constrained estimates of ~5 Pg yr-1 (Kok et al., 2021), but Kok et al. explicitly consider only particles with diameters of less than 20 um.

This incomplete characterization of model emission extends to the analysis of deposition and lifetime. The authors write that "dust is predominantly removed by dry deposition (60-86%) in most models," (line 274) but this statement strongly depends

upon the represented size range in each model. Models with larger maximum particle diameters will remove more of their dust using dry deposition and have shorter particle lifetimes (Figure 8) even among models that represent the physical process of deposition identically.

To be sure, variations of particle-size range among models are imposed upon the authors because the CMIP6 archive records only size-integrated emission, load and other variables. Still the analysis in this article would be much more useful if the authors distinguished the impact of uncertain model representations of emission and deposition physics from the varying ranges of particle size. One (uncertain) way of addressing the effect of size is to plot for each model its emission (or its logarithm) vs. the largest particle size.  I would expect that models with larger particles will generally be associated with larger emission.  This will help untangle (albeit imperfectly) the influence of physics and model size range upon the diversity of emission and deposition.

Finally, in the abstract and conclusions, the authors should note the challenge imposed by the absence of size-resolved emission and deposition in the CMIP6 archive and strongly recommend the addition of this dependency for CMIP7. In the abstract, the authors recommend that future MIPs request "More detailed output" (line 19) without providing an example.

Dust load and dust AOD are also subject to this limitation, but to a lesser degree because the larger particles (that might cause the greatest discrepancies in model emission and deposition) make smaller contributions to the former variables.

We thank the reviewer for their very constructive discussions here. As pointed out by the reviewer, all CMIP6 models output size-integrated dust variables, making it challenging to investigate the links between diversities in dust particle sizes and those in dust processes. To stress this particular issue while trying the method the reviewer suggested, we have taken the following actions:

1) We have clarified that the observationally-constrained dust emission estimate is only for $PM_{20}$ particles by "and the recent observationally-constrained estimates of ~5 Pg yr$^{-1}$ for $PM_{20}$ dust particles".

2) We have produced the scatterplots of the global total budgets of dust emission, deposition, lifetime and DOD against the maximum dust particle size represented in each model and reanalysis dataset (Figure R1 below). Note that only 9 models are plotted here because other models either have prescribed bulk dust concentrations or use a modal scheme, whereby the maximum diameter represented is not clear cut. As can be seen and expected, the results are very noisy, and do not show any conclusive links between dust sizes and dust budgets. For dust deposition, we also plotted dry deposition and wet deposition separately, but got similar and noisy results. Given this, we unfortunately have to say there is no clear relationship between maximum dust size represented and global model emission, lifetime or deposition across models regarding this point.

3) In the abstract, we highlight the need of size-resolved model outputs for future purpose; "More detailed output, and dust size-resolved variables in particular, relating to the dust cycle in future intercomparison projects are needed to enable better constraints of global dust cycles, and enable the potential identification of observationally-constrained links between dust cycles and optical properties."

4) In Sect. 4, we highlight this issue by the following: "One particular issue is that different models have various assumptions on dust particle size range. This means that diversities in dust sizes, on top of those in model physical processes, add uncertainties to models' simulated dust processes ranging from emission and deposition to mas loading, lifetime and DOD. We therefore recommend more detailed output relating to the dust cycle in future model intercomparison projects. For example, size-resolved dust cycle variables, dust loading and transport at different altitudes."

[Figure]

Figure R1: Scatterplots of global dust (a) total emission, (b) total deposition, (c) lifetime, and (d) mean DOD versus maximum dust particle diameter in models and reanalysis datasets. NB the logarithm scale of the x-axis.

Minor Comments
42 "as they become a larger fraction of the total aerosol burden" We expect that air-quality regulations will reduce the concentrations and impacts of anthropogenic aerosols, but whether the global dust load becomes larger and thus "has a greater role in shaping future climate variability" is uncertain.
We agree with the reviewer about the uncertainty related to future global dust emissions. However, our intention here was to point out that the *relative* burden of natural aerosols will become greater as anthropogenic aerosols decline. We have revised this sentence for enhanced clarity and quote it here: "Also, in the context of global efforts to mitigate anthropogenic aerosol and precursor emissions, natural aerosols like dust will potentially form a relatively greater and yet uncertain contribution to global aerosol concentrations in shaping future climate variability."

57 "while their year to year changes were poorly constrained compared to observations" This was an ill-posed test. Evan et al. (2014) compared interannual dust variations to CMIP5 twentieth century historical simulations that were initialized in 1850. These models cannot be expected to reproduce observed interannual dust variability any more than the models can be expected to successfully predict the weather on any given day in the late-twentieth century.
We agree that in these model simulations, the precise weather (or dust) patterns on a particular day, month or year would not be expected to be reproduced. However, the year-to-year variability would be expected to resemble that of reality if the models

were performing well. Therefore, we retain this sentence since this work contributes important findings to the field of dust research relating to the present study.

59 "satellite-observed and CMIP5 models' simulated decadal variabilities of dust emissions..." Satellites do not observe dust emission.
We thank the reviewer for their careful read. We have reworded satellite-observed into observed

68 "featured amplified uncertainties" How is uncertainty defined here?
The uncertainties here refer to the inter-model diversities in global total budget of dust emission and deposition. We have reworded 'uncertainties' into 'model diversities.'

110 "the intermediate horizontal resolution" Please specify this resolution explicitly.
The resolution is 1.25°x1.875°, which is now explicitly stated.

154 "we made leaf area index and soil moisture redundant" Do you mean that you deleted them from your regression model?
Yes – after initial statistical analysis, we decided to delete them from the regressions because they bear similar information to other variables. The text now reads "However, we deleted leaf area index and soil moisture from the regression after the variance inflation factor analysis which indicates these two variables bear similar information to others included in the regression (not shown)."

185 "dust emission hotspots" What is meant by 'hotspots' in this sentence?
We have reworded this into "major desert dust source regions"

199 How is "North China" defined?
North China refers to the Chinese desert region (i.e., the Gobi and Taklamakan deserts). We have added a new Figure 1 (copied below) to show the definition of regions used in this study.

[Figure]

**Figure 1**: The CMIP6 AMIP MEM-simulated 2005-2014 annual mean dust emission (g m$^{-2}$ yr$^{-1}$) climatology overlaid by boxes used to define major dust emission source regions. The coloured symbols denote groupings of observations by different regions following Kok et. al., (2021).

205 "CMIP6 models also feature diversities in the global surface area of dust emissions" This is an interesting metric!
Thanks!

226 "MEM and most individual models are much larger (up to 10 times) than those in CAMS and MERRA2" Can you estimate how much of this difference is due to contrasting choices of particle size range by each model?
Please refer to our replies to reviewer #2's major comment #2

235 "Surface wind speed is shown as the dominant driver of dust emissions in all the models and CAMS." Here, it should be noted that the regression model is based upon monthly mean winds while a disproportionate fraction of emission comes from strong winds on shorter time scales that may not always be well-correlated with monthly means.
We totally agree with the reviewer on this point. However, we were not able to test the regression with different timescales due to the lack of high-resolution model output. Nevertheless, we have added te following at the end of Sect. 2.4 to comment on this: "Note that here we use monthly data to feed the regression, while strong winds at shorter time scales may account for disproportionally more dust emissions. However, we were not able to test it due to the lack of high-resolution model outputs."

Figure 3: add a vertical scale to each panel?
Added!

305 "This highlights the inconsistent behaviour of CMIP6 models in simulating the optical depth of different aerosol species." This is an interesting result, but doesn't this inconsistency call into question the derivation of DOD from MIDAS retrievals combined with the fraction of dust from a single model (in this case MERRA2)?
Yes – we agree with the reviewer, and think this is also a great point to make here. We added a sentence following this: "This highlights the inconsistent behaviour of CMIP6 models in simulating the optical depth of different aerosol species. However, it may also question the reliability of the MIDAS retrievals which are based upon MERRA2"

309 Here, you should add the Ridley et al. ACP 2016 observationally constrained estimate of DOD = 0.03 +/- 0.01 for PM20 dust.
Thanks – we have now referred to this study in the revised version.

310 "significant biases in the MEM-simulated DOD magnitudes at regional scales." Do these biases really reflect unrealistic physics of some of the models? How much of this bais is due to the reanalyses using the Ginous JGR 2001 topographic erodibility map rather than the other (physically reasonable) maps used by some models?
We thank the reviewer for their comment.
Regional differences in modelled DOD will occur due to a variety of factors, of which the emission scheme is one of many. Separating emissions and DOD according to the erodibility map applied is outside of the scope of the work addressed by this study.

Figure 9e, f and g.  Please label the axes on the plot.
Thanks for the suggestions, and done in the revised version

351 typo "Out key findings are"
Corrected!

388 "more detailed output relating" Please make specific recommendations
here.  (e.g. add size-resolved variables)
We thank the reviewer for the great comment. We have now expanded this sentence
into "We therefore recommend more detailed output relating to the dust cycle in
future model intercomparison projects, such as, size-resolved dust cycle variables,
and three-dimensional dust burdens, and dust aerosol optical properties."